# LLaVA-MoD: Making LLaVA Tiny via MoE-Knowledge Distillation

**Fangxun Shu**[1][*]  **Yue Liao**[2][*]  **Lei Zhang**[3][†]  **Le Zhuo**[2][†]  **Chenning Xu**[1][†]
**Guanghao Zhang**[1]  **Haonan Shi**[1]  **Long Chen**[1]  **Tao Zhong**[1]  **Wanggui He**[1]
**Siming Fu**[1]  **Haoyuan Li**[1]  **Zhelun Yu**[1]  **Si Liu**[4][‡]  **Hongsheng Li**[2][‡]  **Hao Jiang**[1][‡]
[1]Alibaba Group  [2]The Chinese University of Hong Kong
[3]University of California, San Diego  [4]Beihang University

## Abstract

We introduce LLaVA-MoD, a novel framework designed to enable the efficient training of small-scale Multimodal Language Models ($s$-MLLM) distilling knowledge from large-scale MLLM ($l$-MLLM). Our approach tackles two fundamental challenges in MLLM distillation. First, we optimize the network structure of $s$-MLLM by integrating a sparse Mixture of Experts (MoE) architecture into the language model, striking a balance between computational efficiency and model capability. Second, we propose a progressive knowledge transfer strategy for comprehensive knowledge transfer. This strategy begins with mimic distillation, where we minimize the Kullback-Leibler (KL) divergence between output distributions to enable $s$-MLLM to emulate $l$-MLLM's understanding. Following this, we introduce preference distillation via Preference Optimization (PO), where the key lies in treating $l$-MLLM as the reference model. During this phase, the $s$-MLLM's ability to discriminate between superior and inferior examples is significantly enhanced beyond $l$-MLLM, leading to a better $s$-MLLM that surpasses $l$-MLLM, particularly in hallucination benchmarks. Extensive experiments demonstrate that LLaVA-MoD surpasses existing works across various benchmarks while maintaining minimal activated parameters and low computational costs. Remarkably, LLaVA-MoD-2B surpasses Qwen-VL-Chat-7B with an average gain of 8.8%, using merely 0.3% of the training data and 23% trainable parameters. The results underscore LLaVA-MoD's capability to effectively distill comprehensive knowledge from its teacher model, paving the way for the development of efficient MLLMs. The code is available at https://github.com/shufangxun/LLaVA-MoD.

## 1 Introduction

Multimodal Large Language Models (MLLMs) (Bai et al., 2023b; Liu et al., 2024; Lin et al., 2024b; Chen et al., 2023b; Shu et al., 2023; Lu et al., 2024) have demonstrated promising performance in multimodal tasks by integrating visual encoders with Large Language Models (LLMs) (Achiam et al., 2023; Bai et al., 2023a; Team et al., 2023; Touvron et al., 2023). However, the considerable size of these models and their reliance on vast training data pose significant computational challenges. For instance, the largest version of LLaVA-NeXT utilizes the Qwen-1.5-110B model, requiring 128 H800 GPUs for 18 hours of training. Furthermore, the extensive number of parameters necessitates advanced hardware, resulting in slow inference speeds that complicate real-world deployment, particularly on mobile devices. Consequently, it is essential to explore small-scale MLLMs (s-MLLMs) that strike a balance between performance and efficiency.

Previous works on s-MLLM (Zhou et al., 2024a; Yuan et al., 2023; Shao et al., 2024; He et al., 2024; Chu et al., 2023; Yao et al., 2024) have focused on high-quality data collection and filtering protocols. While effective, they are inherently limited by the model capacity. Knowledge Distillation

---

[*]Equal Contribution
[†]Core Member
[‡]Corresponding Author

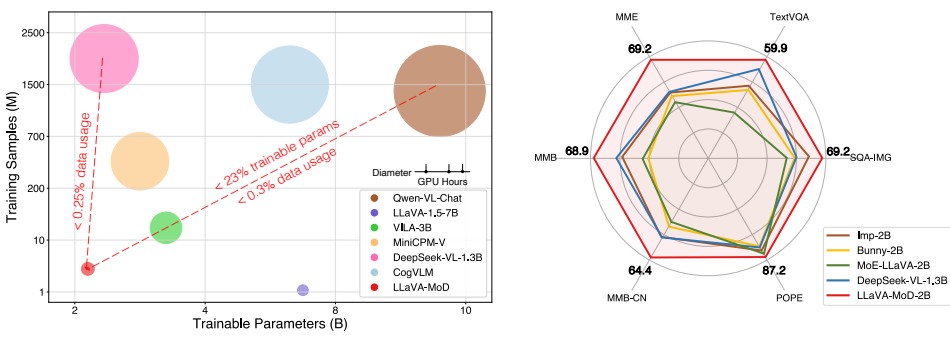

(a) Comparison of training samples, parameters, and GPU hours

(b) Comparison of performance various tasks

Figure 1: Comparisons of training cost and performance. LLaVA-MoD achieves comparable performance with advanced MLLMs using significantly lower training costs while outperforming current small-scale MLLMs by a large margin.

(KD) from large-scale MLLM (*l*-MLLM) offers a promising yet unexplored strategy for enhancing *s*-MLLM performance. By aligning small models' output distributions with large models, KD allows *s*-MLLM to leverage the rich knowledge embedded in *l*-MLLM. We explore a distillation framework for MLLM and consider two primary challenges. The first challenge is to design a lightweight architecture for *s*-MLLM while still maintaining strong learning and expressive capabilities to effectively absorb complex knowledge from *l*-MLLM. The second challenge is to ensure an effective transfer of complex, multi-task knowledge from *l*-MLLM to *s*-MLLM. We present **LLaVA-MoD**[1] to address the challenges through Mixture-of-Expert (MoE) and Knowledge Distillation.

To address the first challenge, one straightforward approach is to create a small model by simply scaling down *l*-MLLM. However, this direct reduction in size can significantly diminish the model's expressive capabilities, resulting in a decline in performance for handling complex multi-modal tasks. Drawing inspiration from the recent success of MoE (Dai et al., 2024; Jiang et al., 2024) in language modeling, we design an MoE structure into *s*-MLLM to balance scale reduction while maintaining the ability to capture complex multimodal knowledge through sparsely activated experts during distillation. Specifically, we equip *s*-MLLM with multiple feedforward networks (FFNs) and a linear gate within the LLM block. Each FFN serves as an expert to capture fine-grained knowledge from *l*-MLLM, while the gate dynamically selects the top-$k$ experts to identify the optimal knowledge transfer pathway and maintain the training and inference efficiency.

To address the second challenge, we propose a progressive distillation strategy. The process begins by aligning the vision encoder with the LLM using a learnable adapter, thereby initializing a dense *s*-MLLM. Subsequently, we employ two consecutive distillation stages, where *s*-MLLM evolves from mimicking and approximating *l*-MLLM to ultimately surpassing it: ***Mimic Distillation.*** This stage is divided into two steps, *i.e.*, dense-to-dense (D2D) and dense-to-sparse (D2S). The rationale for this two-step process is to facilitate the transfer of complex knowledge, where the first step targets general knowledge to build a solid foundation, while the second step targets specialized knowledge to handle multi-task tasks. In D2D, we employ standard KD loss to align the output logits distribution between the initialized dense *s*-MLLM and *l*-MLLM, using general captioning and conversation datasets. Next, in D2S, *s*-MLLM is first transformed from dense to sparse. We then employ the standard KD loss and LM loss to distill the MoE block, using the complex multi-task datasets. ***Preference Distillation.*** In this stage, *l*-MLLM provides knowledge regarding what constitutes "good" and "bad" samples, establishing a foundational reference for the student model. The *s*-MLLM leverages this knowledge to adjust its probability distribution, ensuring that good samples have a higher probability than those from *l*-MLLM, while bad samples are assigned a lower probability. This process enhances the *s*-MLLM's ability to mitigate hallucinations by improving its judgment capabilities beyond those of *l*-MLLM.

LLaVA-MoD exhibits impressive performance on various multimodal benchmarks while maintaining minimal activated parameters and low computational resources. As illustrated in Figure 1, LLaVA-MoD-2B exceeds Qwen-VL-Chat-7B by an average of 8.8% on these benchmarks, uti-

---

[1]MoD denotes Mixture-of-Expert Knowledge Distillation for MLLMs

lizing only 0.3% of the training data and 23% trainable parameters. Furthermore, it matches the performance of RLHF-based methods with 7B and 13B parameters on several hallucination benchmarks. Specifically, LLaVA-MoD-2B surpasses RLHF-V (Yu et al., 2024a) by 8.2% in response-level hallucination rate and by 21.3% in mention-level hallucination rate on the Object HalBench. The impressive results demonstrate the effectiveness of our distillation framework in transferring knowledge from *l*-MLLM to *s*-MLLM.

## 2 RELATED WORK

**Multimodal Large Language Models.** LLMs have greatly advanced the field of natural language processing. Connecting visual information to LLM is crucial for promoting the unified comprehension of vision and language. BLIP-2 (Li et al., 2023b) adds additional intermediary structures to adapt visual features to LLM. Flamingo (Alayrac et al., 2022) incorporates additional cross-attention modules into LLM to handle arbitrarily interleaved multimodal sequences. Recently, models like LLaVA (Liu et al., 2024) and MiniGPT-4 (Zhu et al., 2023) have tried to enhance the models' instruction-following ability through visual instruction tuning. In addition, some works focus on a stronger vision encoder (Chen et al., 2023b; Li et al., 2024) or more powerful fine-grained visual understanding capabilities (Bai et al., 2023b; Wang et al., 2024). Unlike these approaches, our method does not need to significantly enlarge the model. We aim to combine distillation and MoE to improve computational and storage efficiency while maintaining advanced performance.

**Knowledge Distillation.** The enormous sizes and high inference costs of LLMs limit their application in low-resource environments. Knowledge distillation (Hinton et al., 2015) uses a large model as the teacher to transfer its advanced knowledge to a smaller student model, which plays a critical role in compressing model size and enables smaller models to self-improve. MiniLLM (Gu et al., 2023) minimizes the reverse KL divergence to prevent students from overestimating low-probability regions in the teacher distribution, while GKD (Tan et al., 2023) introduces generalized knowledge distillation and promotes the integration of distillation with RLHF. Additionally, some works adopt context distillation (Huang et al., 2022) and CoT (Chain-of-Thought) distillation (Mukherjee et al., 2023; Li et al., 2022; Ho et al., 2022) to enhance specific skills of small models. Our approach innovatively distills the knowledge of MLLM into a smaller sparse MoE architecture, significantly enhancing the multimodal processing capabilities of small models at a minimal cost.

**Mixture-of-Experts** The mixture of experts (MoE) architecture, introduced by (Jacobs et al., 1991), enhances performance by leveraging independent experts to handle diverse samples. In transformer-based architecture, the feed-forward neural network (FFN) layers serve as the experts, sparsely activated through a gating strategy (Lepikhin et al., 2020; Fedus et al., 2022). This design effectively enhances model capacity while keeping computational overhead low. Additionally, the sparse upcycling method (Komatsuzaki et al., 2022) which initializes experts using those from a well-trained dense model is proposed to further mitigate training costs. MoE has shown great potential not only in language models (Jiang et al., 2024; Dai et al., 2024) but also in vision models (Riquelme et al., 2021) and vision-language models (Lin et al., 2024a; Shen et al., 2023). Our approach integrates MoE with knowledge distillation techniques to provide stronger signals for sparse training, which remarkably decreases the training cost associated with sparse models.

## 3 METHOD

We introduce LLaVA-MoD, a novel framework for building efficient *s*-MLLM using mixture-of-experts (MoE) and knowledge distillation. Our framework consists of two main components: *(a). Architecture Design of* s-*MLLM*: As shown in Fig. 3, we design a sparse *s*-MLLM with MoE, enhancing the ability to acquire specialized expert knowledge while maintaining training and inference efficiency. (b). *Distillation Mechanism*: We design a progressive distillation mechanism as shown in Fig. 2 to transfer knowledge from *l*-MLLM to sparse *s*-MLLM. This process involves two stages: mimic distillation followed by preference distillation.

### 3.1 ARCHITECTURE DESIGN OF SPARSE *s*-MLLM

In this section, we describe the architecture design of our sparse *s*-MLLM, which serves as the student model in the distillation process.

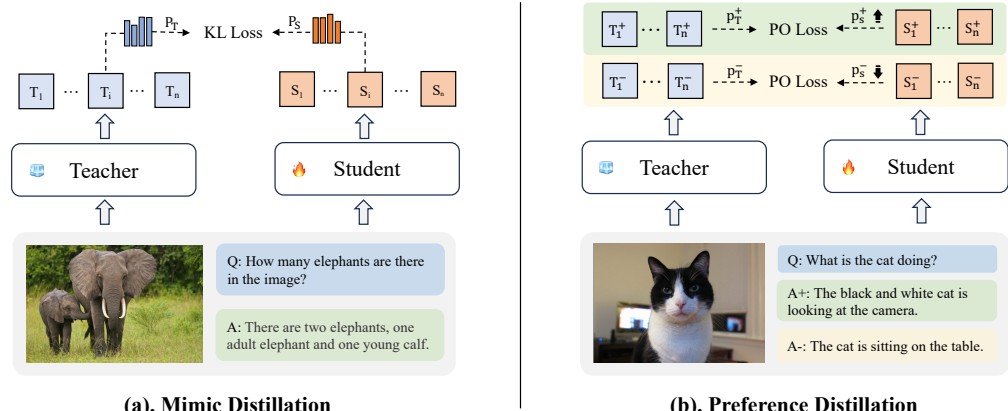

(a). Mimic Distillation        (b). Preference Distillation

Figure 2: Progressive Distillation of LLaVA-MoD. (a). Mimic Distillation: Aligning the student's response probabilities $p_S$ with those of the teacher $p_T$, via the Kullback–Leibler (KL) loss. (b). Preference Distillation: Increasing the student's positive response probabilities $p_S^+$ to surpass those of the teacher $p_T^+$, while decreasing the student's negative response probabilities $p_S^-$ to fall below those of the teacher $p_T^-$, via the Preference-Optimization (PO) loss.

**s-MLLM Definition.** As illustrated in Fig. 3, the basic architecture of *s*-MLLM consists of three primary components: a vision encoder, a large language model (LLM), and a vision-language (VL) adaptor. Given a multimodal instruction conversation $(x, y)$, we define our *s*-MLLM to process response $y$ as follows:

$$y = \text{LLM}_\phi(\text{Proj}_\omega(\text{ViT}_\chi(x_v)), x_i), \tag{1}$$

where $x_v$ is the input image, and $x_i$ is the text instruction. The input image is resized to $336 \times 336$ and patched into 576 image tokens, each of size $14 \times 14$. $\text{ViT}_\chi$ is the CLIP vision encoder with parameters $\chi$ which first extracts image features from $x_v$. $\text{Proj}_\omega$ is the vision-language adaptor with parameters $\omega$, serving as the vision tokenizer to align the image features with the word embedding space. $\text{LLM}_\phi$ is the large language model with parameters $\phi$, which produces the response $y$ based on the multimodal tokens of $x = [x_v, x_i]$.

**Sparsify *s*-MLLM.** The principle of building our *s*-MLLM is downsizing the LLM while leaving the vision encoder and vision-language adaptor unmodified. To achieve this downsizing goal, we sparsify the dense *s*-MLLM by incorporating an MoE architecture. Specifically, Fig. 3 illustrates the process, where we apply the sparse upcycling technique (Komatsuzaki et al., 2022) to replicate $N$ feedforward networks (FFNs) as the expert modules. Additionally, we introduce a linear layer as the router, which dynamically activates the appropriate experts by predicting the probability of expert assignment. Given each token $x$ in the sequence, we first compute the routing value of $N$ experts:

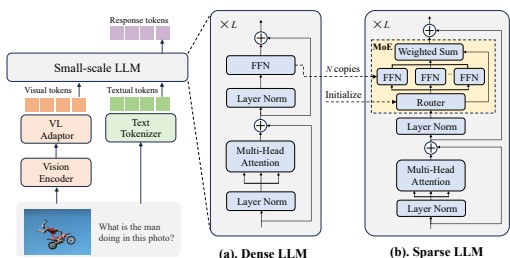

Figure 3: Sparsification of *s*-MLLM. The VL Adaptor and Vision Encoder remain unchanged, while the LLM is upcycled from dense to sparse.

$$r = \text{Softmax}(x \cdot W_r), \tag{2}$$

where $W_r$ denotes the weight matrix of the router and each element $r_i$ in $r$ represents the probability of activating the $i$-th expert. After that, we apply the Top-$k$ strategy to determine the activated experts with the highest $k$ routing values:

$$\tilde{r} = \text{Top-}k(r) = \begin{cases} r_i, & \text{if } i \in k \\ 0, & \text{otherwise} \end{cases} \tag{3}$$

where the routing values for the inactivated experts are set to zero, effectively excluding them from contributing to the final output. The output $y$ is calculated by aggregating the contributions of the

activated experts, weighted by their corresponding routing values:

$$y = \sum_{i=1}^{N} \tilde{r} \cdot E_i(x). \tag{4}$$

## 3.2 PROGRESSIVE DISTILLATION

Our progressive distillation consists of two distinct stages, *i.e.*, mimic distillation (Fig. 2 (a)) and preference distillation (Fig. 2 (b)). In the mimic distillation stage, the *s*-MLLM $\pi_S$ imitates the general and specific knowledge from the *l*-MLLM $\pi_T$. In the preference distillation stage, $\pi_S$ gains the preference knowledge of $\pi_T$ to further refine its output and reduce hallucinations. Both $\pi_S$ and $\pi_T$ are from the same LLM family. This ensures a consistent vocabulary space, which is essential for accurate imitation.

**Initialization.** Before distillation, we first align the vision encoder with the LLM via a learnable adapter, aiming to obtain a well-initialized dense version of $\pi_S$. The LLM$_\phi$ and ViT$_\chi$ are maintained frozen since their pre-trained parameters have already captured rich visual and language knowledge. Only Proj$_\omega$ is optimized to bridge the gap between the vision and language domain. For the initialization, we utilize common image-caption pairs from a widely used and curated dataset, which covers a diverse range of subjects and visual entities. The training objective is to minimize the cross-entropy of the generated tokens. The objective function is:

$$\mathcal{L}_{\text{Init}}(\pi_S) = -\mathbb{E}_{(y_k|y_{<k},x)\sim\pi_S}\left[\log \pi_S(y_k \mid y_{<k}, x)\right], \tag{5}$$

where $\pi_S(y_k \mid y_{<k}, x)$ represents the probability of the predicted token $y_k$ conditioned on $x$ and the previous tokens $y_{<k} = (y_1, y_2, \ldots, y_{k-1})$.

**Mimic Distillation.** We decompose the comprehensive knowledge within $\pi_T$ into general and specialized aspects to address the challenges posed by their structural differences, which can complicate simultaneous learning. Subsequently, we conduct a general-to-specialized mimic distillation, which consists of two steps: dense-to-dense (D2D) and dense-to-sparse (D2S) distillation, to transfer the knowledge to $\pi_S$. This two-step approach balances the transfer of general and specialized knowledge through the progressive distillation, thereby enhancing overall performance. As illustrated in Fig. 3, we utilize the dense structure of $\pi_S$ during D2D to acquire general knowledge and transform it into a sparse structure during D2S to acquire complex specialized knowledge. Throughout both stages, $\pi_T$ remains unchanged.

***a). Dense-to-Dense Distillation.*** In this stage, we aim to replicate the general knowledge of *l*-MLLM. Acquiring general knowledge first is crucial, as it establishes a broad foundation across various scenarios, enabling *s*-MLLM to develop a basic framework before advancing to multiple specialized tasks. To achieve this, we maintain ViT$_\chi$ frozen, and jointly optimize LLM$_\phi$ and Proj$_\omega$, with trainable parameters $\theta = \{\omega, \phi\}$. We leverage common image-caption pairs and conversation datasets. The training objective is to minimize the Kullback-Leibler divergence (KLD) between the output logits of *s*-MLLM and *l*-MLLM. The objective function is:

$$\mathcal{L}_{\text{D2D}}(\pi_S; \pi_T) = -\mathbb{E}_{(x,y_k)\sim\pi_T}\left[\log \frac{\pi_T(y_k \mid y_{<k}, x)}{\pi_S(y_k \mid y_{<k}, x)}\right], \tag{6}$$

where $V$ denotes the vocabulary, while $\pi_S(y_k \mid y_{<k}, x)$ and $\pi_T(y_k \mid y_{<k}, x)$ denote the probability of the predicted tokens for *s*-MLLM and *l*-MLLM.

***b). Dense-to-Sparse Distillation.*** In this stage, our focus shifts to transfer the specialized knowledge of *l*-MLLM into *s*-MLLM to obtain advanced capabilities and achieve superior performance in complex tasks. However, directly learning this knowledge in the dense form of *s*-MLLM could be insufficient due to the diverse knowledge structure of different tasks. Therefore, we sparsify the dense *s*-MLLM by introducing multiple experts. As detailed in Section 3.1, we replicate $N$ feedforward networks (FFNs) within LLM$_\phi$ and add an MLP layer as a router, forming the experts with parameters $\phi_e$. This sparse architecture enables *s*-MLLM to activate the most relevant experts based on different inputs selectively, offering significant advantages in emulating the specialized knowledge of *l*-MLLM. For training, we leverage multi-task data, updating only the experts and the adaptor. We employ a Top-$k$ routing strategy to select the experts. The trainable parameters are $\theta = \{\omega, \phi_e\}$. Similar to the previous stage, we adopt the KLD as the training objective. Additionally, we include the standard next-token prediction objective, *i.e.*, minimizing the cross-entropy loss

of generated tokens by *s*-MLLM, to inject supervision from ground truth data which can reduce the existing bias in *l*-MLLM. The objective function is:

$$\mathcal{L}_{\text{D2S}}(\pi_{\text{S}}; \pi_{\text{T}}) = -\mathbb{E}_{(x,y_k)\sim\pi_{\text{S}}}\left[\log \pi_{\text{S}}(y_k \mid y_{<k}, x)\right] - \mathbb{E}_{(x,y_k)\sim\pi_{\text{T}}}\left[\log \frac{\pi_{\text{T}}(y_k \mid y_{<k}, x)}{\pi_{\text{S}}(y_k \mid y_{<k}, x)}\right]. \tag{7}$$

**Preference Distillation.** In this stage, our goal is to distill the preference knowledge from *l*-MLLM to guide *s*-MLLM towards generating not only accurate but also reasonable responses, which is crucial in reducing hallucinations. For the training process, we effectively use preference data, which comprises meticulously paired positive responses $y^+$ and negative responses $y^-$ for the identical prompt $x$. Our preference distillation strategy is inspired by recent advancements in Preference Optimization (PO) (Rafailov et al., 2024; Ethayarajh et al., 2024), which bypasses the need for training a reward model by directly training on an offline preference dataset. Our key insight is to treat *l*-MLLM as the reference model to provide insights on what constitutes "good" and "bad", thereby establishing a fundamental reference for *s*-MLLM.

Specifically, the training objective is to optimize *s*-MLLM to assign higher probabilities to positive responses and lower probabilities to negative ones compared to *l*-MLLM. This training process involves two key optimization aspects: First, *s*-MLLM aims to align with the teacher model in distinguishing positive from negative responses. Second, *s*-MLLM seeks to surpass *l*-MLLM by assigning higher probabilities to positive responses and lower probabilities to negative responses. We only train the experts and adaptor in *s*-MLLM and employ a Top-$k$ routing strategy to select the experts. The trainable parameters are $\theta = \{\omega, \phi_e\}$. The objective function is:

$$\mathcal{L}_{\text{PD}}(\pi_{\text{S}}; \pi_{\text{T}}) = -\mathbb{E}_{(x,y^+,y^-)\sim\mathcal{D}}\left[\log \sigma \left(\beta \log \frac{\pi_{\text{S}}(y^+ \mid x)}{\pi_{\text{T}}(y^+ \mid x)} - \beta \log \frac{\pi_{\text{S}}(y^- \mid x)}{\pi_{\text{T}}(y^- \mid x)}\right)\right], \tag{8}$$

where $\pi_{\text{S}}(y^+ \mid x)$ and $\pi_{\text{S}}(y^- \mid x)$ denote the probabilities of positive and negative responses for *s*-MLLM, and $\pi_{\text{T}}(y^+ \mid x)$ and $\pi_{\text{T}}(y^- \mid x)$ denote the same for *l*-MLLM.

## 4 EXPERIMENTS

### 4.1 EXPERIMENTAL SETTINGS

**Implementation Details.** We employ the "ViT-MLP-LLM" architecture to demonstrate the effectiveness of LLaVA-MoD. A pre-trained CLIP-ViT-L/14 is utilized as the vision encoder and a 2-layer MLP is utilized as the adaptor. Qwen-1.5/2 with different sizes are utilized as the LLM for *l*-MLLM and *s*-MLLM. Specifically, *l*-MLLM is configured with 7B parameters, while *s*-MLLM is configured with 1.8B and 0.5B parameters. The performance of *l*-MLLM on multimodal benchmarks is presented in Tab. 1. We use the same series of LLMs for distillation, *i.e.* employing Qwen-1.5 7B to distill Qwen-1.5 1.8B and Qwen-1.5 0.5B. Each stage uses distinct training configurations. The detailed training strategy and hyperparameter are illustrated in Appendix A.1.

**Training Datasets.** The training data consists of 5M samples from the open-source datasets, with each training stage utilizing a distinct dataset. During Initialization, 0.6M general captioning samples are used to bridge the gap between visual and language modalities. In mimic distillation, 2.4M general captioning and conversation samples are used to distill general knowledge from *l*-MLLM, and 1.4M multi-tasks data, including VQA, documents, science, and OCR, are used to distill specialized knowledge from *l*-MLLM. For preference distillation, 80K preference data samples are used to transfer preference knowledge from *l*-MLLM. The detailed dataset of each training stage is illustrated in Appendix A.2.

**Evaluation Benchmarks.** We conduct experiments on MME (Fu et al., 2023), MMB (Liu et al., 2023c), and MMB$^{\text{CN}}$. Each encompasses various sub-tasks, enabling comprehensive evaluation of multimodal understanding and reasoning capabilities. Additionally, we carry out experiments across a broad spectrum of VQA tasks, which include general VQA, text-oriented VQA, and science VQA. Specifically, for general VQA tasks, we use VizWiz (Gurari et al., 2018) and GQA (Hudson & Manning, 2019) to test general visual understanding and relational reasoning. TextVQA (Singh et al., 2019) is employed for text-oriented VQA tasks, focusing on fine-grained visual recognition and understanding of text within images. ScienceQA (Lu et al., 2022b) is utilized to measure scientific knowledge. Moreover, we conduct experiments on several hallucination benchmarks such as POPE (Li et al., 2023c), Object HalBench (Yu et al., 2024a), MMHal-Bench (Sun et al., 2023).

Table 1: Comparison with state-of-the-art MLLMs on the commonly-used multimodal benchmarks for MLLMs. #Sample: Training data sample. #Param: Trainable parameters. SQA$^I$: ScienceQA test, VQA$^T$: TextVQA val, MME: MME Benchmark, normalized to percentage, MMB: MMBench dev, MMB$^{CN}$: MMBench-Chinese dev. The best result for model sizes 1B/2B is shown in bold, and the second-best result is underlined. Our LLaVA-MoD achieves the best average result for both.

| Method | LLM | #Sample | #Param | GQA | VisWiz | SQA$^I$ | VQA$^T$ | MME | MMB | MMB$^{CN}$ | AVG |
|---|---|---|---|---|---|---|---|---|---|---|---|
| Teacher MLLM | Qwen-1.5-7B | 5M | ~7B | 62.3 | 40.7 | 70.9 | 55.3 | 72.1 | 68.5 | 64.9 | 62.1 |
| Teacher MLLM | Qwen-2-7B | 5M | | 62.5 | 37.9 | 73.2 | 57.2 | 78.0 | 71.6 | 71.1 | 64.5 |
| BLIP-2 | Vicuna-13B | 129M | | 41.0 | 19.6 | 61.0 | 42.5 | 64.7 | - | - | - |
| VILA-7B | LLaMA-7B | 50M | | 62.3 | 57.8 | 68.2 | 64.4 | 76.7 | 68.9 | 61.7 | 65.7 |
| CogVLM | Vicuna-7B | 1500M | | 64.9 | - | 65.6 | 78.2 | 71.8 | 63.7 | 53.8 | - |
| InstructBLIP | Vicuna-13B | 130M | ≥7B | 49.5 | 33.4 | 63.1 | 50.7 | 60.6 | - | - | - |
| Qwen-VL-Chat | Qwen-7B | 1450M | | 57.5 | 38.9 | 68.2 | 61.5 | 74.4 | 60.6 | 56.7 | 56.7 |
| Deepseek-VL-7B | DLLM-7B | 2000M | | 61.3 | 49.9 | 74.0 | 64.7 | 73.4 | 74.1 | 72.8 | 67.2 |
| LLaVA-1.5-7B | Vicuna-1.5-7B | 1.2M | | 62.0 | 50.0 | 66.8 | 58.2 | 75.5 | 64.3 | 58.3 | 62.1 |
| LLaVA-NeXT | Vicuna-1.5-13B | 1.3M | | 65.4 | 60.5 | 73.6 | 67.1 | 78.7 | 70.0 | 64.4 | 68.5 |
| Imp-3B | Phi-2-2.7B | 1.6M | | 63.5 | 54.1 | 72.8 | 59.8 | 72.3 | 72.9 | 46.7 | 63.2 |
| Bunny-3B | Phi-2-2.7B | 2.7M | | 62.5 | 43.8 | 70.9 | 56.7 | 74.4 | 68.6 | 37.2 | 59.2 |
| VILA-3B | LLaMA-2.7B | 51M | | 61.5 | 53.5 | 69.0 | 60.4 | 72.1 | 63.4 | 52.7 | 61.8 |
| MobileVLM | MLLaMA-2.7B | 1.3M | | 59.0 | - | 61.0 | 47.5 | 64.4 | 59.6 | - | - |
| MobileVLM$^{v2}$ | MLLaMA-2.7B | 3.6M | ~3B | 61.1 | - | 70.0 | 57.5 | 72.0 | 63.2 | - | - |
| MoE-LLaVA-3B | Phi-2-2.7B | 2.2M | | 61.4 | 43.9 | 68.5 | 51.4 | 71.1 | 65.2 | 41.8 | 57.6 |
| MiniCPM-V | MiniCPM-2.4B | 570M | | 51.5 | 50.5 | 74.4 | 56.6 | 68.9 | 64.0 | 62.7 | 61.2 |
| MiniCPM-V-2 | MiniCPM-2.4B | 570M | | 52.1 | 60.2 | 76.3 | 73.2 | 70.5 | 68.5 | 67.2 | 66.9 |
| Imp-2B | Qwen-1.5-1.8B | 1.6M | | **61.9** | 39.6 | 66.1 | 54.5 | 65.2 | 63.8 | 61.2 | 58.9 |
| Bunny-2B | Qwen-1.5-1.8B | 2.7M | | 59.6 | 34.2 | 64.6 | 53.2 | 65.0 | 59.1 | 58.5 | 56.3 |
| Mini-Gemini-2B | Gemma-2B | 2.7M | | 60.7 | **41.5** | 63.1 | 56.2 | 67.0 | 59.8 | 51.3 | 57.1 |
| MoE-LLaVA-2B | Qwen-1.5-1.8B | 2.2M | ~2B | 61.5 | 32.6 | 63.1 | 48.0 | 64.6 | 59.7 | 57.3 | 55.3 |
| DeepSeek-VL-1.3B | DLLM-1.3B | 2000M | | 59.3 | 36.8 | 64.2 | 58.4 | 65.3 | 64.6 | 61.0 | 58.5 |
| LLaVA-MoD-2B | Qwen-1.5-1.8B | 5M | | 58.7 | 39.2 | 68.0 | 58.5 | 67.6 | 66.3 | 61.9 | 59.9 |
| LLaVA-MoD-2B | Qwen-2-1.5B | 5M | | 58.8 | 40.4 | **69.2** | 59.9 | 69.2 | 68.9 | 64.4 | **61.6** |
| SPHINX-Tiny | TLLaMA-1.1B | 15M | | **58.0** | 49.2 | 21.5 | 57.8 | 63.1 | 56.6 | 37.8 | 49.2 |
| LLaVA-MoD-1B | Qwen-1.5-0.5B | 5M | ~1B | 56.2 | 31.6 | **62.8** | 53.9 | 65.3 | **58.8** | 50.4 | 54.1 |
| LLaVA-MoD-1B | Qwen-2-0.5B | 5M | | 56.6 | 35.1 | 61.1 | 57.1 | **67.0** | 58.7 | **54.1** | **55.7** |

## 4.2 MAIN RESULTS

In this section, we conduct experiments of LLaVA-MoD to highlight its advantages in two aspects: performance and efficiency. For performance, we evaluate comprehension-oriented benchmarks (Tab. 1) and hallucination-oriented benchmarks (Tab. 2). For efficiency, we present a comparison in terms of data samples and model size. The performance is obtained under the same series of LLMs, *i.e.* employing Qwen-1.5 7B to distill Qwen-1.5 1.8B and Qwen-1.5 0.5B.

**Comprehension-Oriented Benchmarks.** As indicated in Tab. 1, LLaVA-MoD achieves SOTA average results among the models of 1B and 2B size on comprehension-oriented benchmarks. The 2B-sized LLaVA-MoD surpasses Mini-Gemini-2B (Li et al., 2024) by 8.1%, while using a lower image resolution (336 vs. 768). The 1B-sized LLaVA-MoD surpasses SPHINX-Tiny (Gao et al., 2024) by 13.2%, using fewer data samples (5M vs. 15M). Furthermore, LLaVA-MoD-2B matches and even surpasses the performance of large-scale MLLMs. The 2B-sized LLaVA-MoD surpasses Qwen-VL-Chat-7B (Bai et al., 2023b) by 8.8% and matches the performance of VILA-3B (Lin et al., 2024b) and MiniCPM-V (Yao et al., 2024). These results highlight that our approach trains small-scale MLLMs effectively by distilling the sparse MoE architecture from large-scale MLLMs.

**Hallucination-Oriented Benchmarks.** As indicated in Tab. 2, LLaVA-MoD shows remarkable performance in mitigating hallucination, even beating its teacher model. It can be attributed to preference distillation from two aspects: Firstly, by assigning a higher probability for the positive response, preference distillation encourages LLaVA-MoD to focus on providing correct and relevant information. Secondly, by assigning a lower probability for the negative response, preference distillation discourages incorrect or unsubstantiated information. By using the teacher model as a reference to adjust response probabilities, preference distillation enables LLaVA-MoD to handle hallucination issues more accurately and reliably, thereby surpassing its teacher model. Moreover, LLaVA-MoD

Table 2: Comparison with state-of-the-art MLLMs on the hallucination benchmarks. We compare LLaVA-MoD ■ with SFT-based works ■ and RLHF-based works ■. Hall: Hallucination Rate Resp: response-level hallucination rate, Ment: mention-level hallucination rate. The best result is in bold, and the second-best result is underlined.

| Model | LLM | #Param | Object HalBench | | POPE | MMHal-Bench | |
| --- | --- | --- | --- | --- | --- | --- | --- |
| | | | Resp ↓ | Ment ↓ | F1 ↑ | Score ↑ | Hall ↓ |
| Teacher MLLM | Qwen-1.5-7B | 7B | 50.1 | 24.8 | 84.9 | 2.60 | 20.7 |
| Teacher MLLM | Qwen-2-7B | 7B | 29.7 | 23.4 | 85.7 | 2.88 | 14.5 |
| Qwen-VL-Chat | Qwen-7B | 9.6B | 40.4 | 20.7 | 74.9 | 2.76 | 38.5 |
| LLaVA-1.5-7B | Vicuna-7B | 7B | 53.6 | 25.2 | 86.1 | 2.36 | 51.0 |
| VCD | Vicuna-1.5-7B | 7B | 48.8 | 24.3 | 84.5 | 2.12 | 54.2 |
| OPERA | Vicuna-1.5-7B | 7B | 45.1 | 22.3 | 85.4 | 2.15 | 54.2 |
| HA-DPO | Vicuna-1.5-7B | 7B | 39.9 | 19.9 | 86.8 | 1.98 | 60.4 |
| POVID | Vicuna-1.5-7B | 7B | 48.1 | 24.4 | 86.3 | 2.08 | 56.2 |
| LLaVA-RLHF | Vicuna-1.5-13B | 13B | 38.1 | 18.9 | 82.7 | 2.02 | 62.5 |
| LURE | Vicuna-1.5-7B | 7B | 27.7 | 17.3 | - | 1.64 | 60.4 |
| RLHF-V | Vicuna-13B | 13B | 12.2 | 7.5 | 86.2 | 2.45 | 51.0 |
| RLAIF-V | Vicuna-1.5-7B | 7B | 8.5 | 4.3 | - | 3.06 | 29.2 |
| MiniCPM-V | MiniCPM-2.4B | 2.8B | 21.6 | 11.5 | 79.5 | 3.70 | 24.9 |
| MiniCPM-V-2 | MiniCPM-2.4B | 2.8B | 14.5 | 7.8 | 86.3 | **4.09** | 18.2 |
| Mini-Gemini-2B | Gemma-2B | 2B | 29.7 | 21.1 | 85.6 | 2.83 | 18.8 |
| Bunny-2B | Qwen-1.5-1.8 | 2.2B | 50.2 | 23.4 | 85.8 | 2.72 | 19.3 |
| LLaVA-MoD-2B | Qwen-1.5-1.8B | 2.2B | 11.4 | 7.2 | 87.0 | 2.76 | 17.2 |
| LLaVA-MoD-2B | Qwen-2-1.5B | 1.9B | **11.2** | **5.9** | **87.2** | 2.91 | **13.8** |

even surpasses recent RLHF-based models (Sun et al., 2023; Zhou et al., 2024b; Huang et al., 2024; Leng et al., 2024; Zhou et al., 2023). On the Object HalBench, it outperforms RLHF-V (Yu et al., 2024a) by 8.2% in response-level hallucination rate and by 21.3% in mention-level hallucination rate. This shows that preference distillation is an effective task in minimizing hallucination.

**Efficiency Comparison.** As indicated in Tab. 18, LLaVA-MoD achieves significant efficiency in both training and inference. Compared to Qwen-VL-Chat-7B, our 2B model achieves 8.8% higher accuracy with only 0.3% of its training data (5M samples) and 23% of activated parameters (2.2B total), while significantly reducing computational costs: on a single A100-80G GPU, it demonstrates 2.5× faster decoding speed, 26% FLOPs, and 38% memory consumption. This efficiency advantage extends to other SOTA models like MiniCPM-V (2.8B parameters), where LLaVA-MoD outperforms using only 1. 6% of training data despite a similar model size. Therefore, LLaVA-MoD establishes a new efficiency frontier for deploying high-performance multimodal models in resource-constrained scenarios.

## 4.3 ABLATION STUDY

In this section, we thoroughly explore the effect of the distillation strategy and model design. Firstly, we investigate the preference distillation on comprehension and hallucination benchmarks. Since our focus is on the training strategy and model design, we carry out subsequent ablation experiments specifically on mimic distillation. We use CLIP-ViT-L/14 as the default vision encoder, and Qwen-1.5-1.8B and Qwen-1.5-7B as the default student and teacher LLMs respectively.

### 4.3.1 IMPACT OF PREFERENCE DISTILLATION

**Preference Distillation Mitigates Hallucination.** We explore the effect of preference distillation on comprehension capability and hallucination mitigation. As shown in Tab. 3 and Tab. 4, preference distillation remarkably reduces the hallucination of *s*-MLLM, while it does not yield consistent improvements in comprehension capability. This observation aligns with that in LLMs (Achiam et al., 2023), where preference optimization tends to prioritize reducing hallucinations, often resulting in a certain degree of decline in comprehension gains.

**KTO Stabilizes Preference Distillation.** We investigate the training stability associated with various preference-optimization losses, specifically employing KTO (Ethayarajh et al., 2024) and DPO (Rafailov et al., 2024). As demonstrated in Appendix B.2, both methods contribute to im-

Table 3: The impact of preference distillation on the comprehension-oriented benchmarks.

| Distillation | GQA | VizWiz | SQA$^I$ | VQA$^T$ | MME | MMB | MMB$^{CN}$ | AVG |
|---|---|---|---|---|---|---|---|---|
| Mimic | 59.3 | 40.0 | 68.4 | 58.7 | 68.4 | 65.1 | 61.5 | 60.2 |
| Mimic+Preference | 58.7 | 39.2 | 68.0 | 58.5 | 66.7 | 66.3 | 61.9 | 59.9 |

Table 4: The impact of preference distillation on the hallucination-oriented benchmarks.

| Distillation | Object HalBench | | POPE | MMHal-Bench | |
|---|---|---|---|---|---|
| | Resp ↓ | Ment ↓ | F1 ↑ | Score ↑ | Hall ↓ |
| Mimic | 39.1 | 22.6 | 86.7 | 2.75 | 17.8 |
| Mimic+Preference | 11.4 | 7.2 | 87.0 | 2.76 | 17.2 |

proved hallucination mitigation; however, KTO exhibits a significant advantage over DPO. This finding suggests that KTO, which categorizes samples as either good or bad, offers a more effective signal for guiding the *s*-MLLM to outperform the *l*-MLLM, in contrast to DPO, which relies on comparing two samples to determine superiority. Additionally, we compare our MoD with existing Knowledge Distillation methods, including KD (Hinton et al., 2015) and GKD (Tan et al., 2023), as detailed in Appendix B.3, to highlight the advantages of our proposed approach.

### 4.3.2 IMPACT OF TRAINING STRATEGY

**KD Facilitates MoE Training.** We explore the effect of knowledge distillation (KD) in MoE training by comparing it with supervised fine-tuning (SFT). The ablation experiments are carried out using the same sparse configuration of E4T2, where four experts are initialized and the top two experts are activated. As shown in Tab. 5, KD surpasses SFT on all benchmarks, achieving an 8.1% average gain over SFT. Additionally, it achieves notable gains in complex multi-task scenarios, such as a +8.2% gain on MMB and a +10.0% gain on MME. These results indicate that KD provides a better optimization signal for effectively training the MoE architecture.

Table 5: Comparison between KD and SFT. The MoE architecture is E4T2.

| Training | GQA | VizWiz | SQA$^I$ | VQA$^T$ | MME | MMB | MMB$^{CN}$ | AVG |
|---|---|---|---|---|---|---|---|---|
| SFT | 58.6 | 31.2 | 66.2 | 55.6 | 63.2 | 59.2 | 56.1 | 55.7 |
| KD | 59.3 | 40.0 | 68.4 | 58.7 | 68.4 | 65.1 | 61.5 | 60.2 |

**General-to-Specialized KD Boosts Performance.** We explore the effect of the general-to-specialized approach in mimic distillation by comparing the proposed **D2D+D2S** with **D2S**. The D2D+D2S involves a two-phase process. Firstly, it distills a dense *s*-MLLM using general data. Subsequently, it transforms the *s*-MLLM from dense to sparse and further distills it using multi-task data. The D2S directly distills the sparse *s*-MLLM using a combined of general and multi-task dataset. Tab. 6 shows that D2D+D2S surpasses D2S with an average gain of 4.0%. The result indicates that D2D+D2S enables effective knowledge transfer, striking a balance between general and specialized competencies. Additionally, D2D+D2S is more computationally efficient, consuming only 62.5% of the GPU hours required by D2S.

Table 6: Comparison between different distillation strategies in mimic distillation.

| Strategy | #GPU | GQA | VizWiz | SQA$^I$ | VQA$^T$ | MME | MMB | MMB$^{CN}$ | AVG |
|---|---|---|---|---|---|---|---|---|---|
| D2S | 1536 | 57.9 | 36.5 | 66.3 | 57.5 | 65.7 | 61.4 | 60.0 | 57.9 |
| D2D+D2S | 960 | 59.3 | 40.0 | 68.4 | 58.7 | 68.4 | 65.1 | 61.5 | 60.2 |

**Focus on Response Distillation Improves Generalization.** We explores the effect of the distillation target by comparing **response** distillation with **response+instruction** distillation. Tab. 7 shows that response distillation surpasses response+instruction distillation with an average gain of 3.1%. The result indicates that focusing on response is more effective for knowledge distillation in autoregressive modeling. A possible reason for this is that mimicking additional instructions can lead to

overfitting the specific instruction patterns in the training data, potentially reducing the generalization to unseen instructions.

Table 7: Comparison between different distilled tokens. **Response** indicates distilling solely on the response tokens. **Response+Instruction** incorporates additional distillation of instruction tokens.

| Distilled Tokens | GQA | VizWiz | SQA$^{\text{I}}$ | VQA$^{\text{T}}$ | MME | MMB | MMB$^{\text{CN}}$ | AVG |
|---|---|---|---|---|---|---|---|---|
| Response + Instruction | 58.5 | 35.0 | 66.8 | 58.3 | 68.5 | 61.8 | 59.0 | 58.4 |
| Response | 59.3 | 40.0 | 68.4 | 58.7 | 68.4 | 65.1 | 61.5 | 60.2 |

### 4.3.3 IMPACT OF MODEL ARCHITECTURE

**Sparse Architecture Facilitates Knowledge Transfer.** We explore the effect of sparse architecture in distillation by comparing it with dense architecture. The configuration is E4T1 where four experts are initialized and the top-1 expert is activated to ensure that the activated parameters are the same as those of the dense architecture. Tab. 8 shows that sparse architecture surpasses dense architecture with an average gain of 3.7%, The superior performance is prominent in complex multi-task benchmarks, such as MME and MMB. The result indicates that sparse architecture leverages MoE to effectively transfer diverse knowledge from $l$-MLLM while maintaining computational efficiency. We also conduct a comprehensive investigation into the performance and computational trade-offs associated with various MoE configurations, including the number of experts, the routing parameter k, and the routing strategies employed in Appendix B.1.

Table 8: Comparison between sparse and dense architecture within the distillation.

| Architecture | GQA | VizWiz | SQA$^{\text{I}}$ | VQA$^{\text{T}}$ | MME | MMB | MMB$^{\text{CN}}$ | AVG |
|---|---|---|---|---|---|---|---|---|
| Dense | 57.6 | 32.7 | 67.3 | 56.8 | 65.2 | 61.8 | 58.2 | 57.1 |
| Sparse | 58.7 | 36.9 | 67.9 | 58.2 | 66.1 | 64.5 | 61.7 | 59.2 |

**Teacher Capacity Matters in Knowledge Distillation.** We explore the effect of the teacher capacity in the distillation. We employ Qwen-1.5-7B as the strong teacher and Qwen-1.5-4B as the weaker one. The MoE configuration is set as E4T2. Tab. 9 shows that a well-suited teacher can boost performance. For Qwen-1.5-1.8B, using a 7B-sized teacher yields an average gain of 2.2% compared to a 4B-sized teacher. However, when the student and teacher have a large capacity disparity, it can hinder effective knowledge transfer. For Qwen-1.5-0.5B, using a 7B-sized teacher yields a marginal average gain compared to a 4B-sized teacher. Utilizing a "middle teacher" with an intermediate capacity can bridge the gap, facilitating smooth knowledge transfer.

Table 9: Comparison between the strong and weak teachers. We set the strong teacher as the LLM is Qwen-1.5-7B and the weak teacher as the LLM is Qwen-1.5-4B.

| Student | Teacher | GQA | VizWiz | SQA$^{\text{I}}$ | VQA$^{\text{T}}$ | MME | MMB | MMB$^{\text{CN}}$ | AVG |
|---|---|---|---|---|---|---|---|---|---|
| Qwen-1.5-0.5B | Qwen-1.5-4B | 56.0 | 25.3 | 64.7 | 53.8 | 63.3 | 62.2 | 50.8 | 53.7 |
| | Qwen-1.5-7B | 56.1 | 29.8 | 63.8 | 54.5 | 64.2 | 58.5 | 49.7 | 53.8 |
| Qwen-1.5-1.8B | Qwen-1.5-4B | 58.7 | 34.6 | 67.9 | 57.7 | 67.6 | 64.9 | 60.7 | 58.9 |
| | Qwen-1.5-7B | 59.3 | 40.0 | 68.4 | 58.7 | 68.4 | 65.1 | 61.5 | 60.2 |

## 5 CONCLUSION

In this paper, we introduce LLaVA-MoD, a novel framework for efficient training of small-scale multimodal language models via knowledge distillation from large-scale ones. It addresses two key challenges in MLLM distillation: enhancing s-MLLM architecture with MoE design for efficiency and expressiveness balance and implementing a progressive knowledge transfer strategy. Extensive experiments show that LLaVA-MoD outperforms existing models with low activation parameters and computational costs. Notably, it outperforms Qwen-VL-Chat-7B by 8.8% with only 2 billion activation parameters, 0.3% training data, and 23% trainable parameters, highlighting its effectiveness in distilling knowledge and driving more efficient MLLM development.

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

## A  IMPLEMENTATION DETAILS

### A.1  TRAINING STRATEGY AND HYPERPARAMETERS

We first freeze the vision encoder and LLM while optimizing the VL adaptor to align image tokens with the word embedding space during initialization. This stage employs a cross-entropy loss with a batch size of 512 and a learning rate of 1e-4. In mimic distillation, the vision encoder remains frozen, while the LLM and VL adaptor are co-optimized to distill general knowledge from *l*-MLLM in dense-to-dense distillation. Then, the FFN in the LLM is first transformed into a sparse architecture with a mixture of FFNs co-optimized with the VL adaptor to distill specialized knowledge from *l*-MLLM in dense-to-sparse distillation. This stage employs a KL-divergence loss, with cross-entropy loss added for dense-to-sparse distillation. The batch size is 256, and the learning rate is

adjusted to 2e-5. In preference distillation, the model inherits from the mimic distillation. The vision encoder remains frozen, and a mixture of FFN experts is co-optimized with the VL adaptor to distill preference knowledge from the teacher MLLM. This stage employs a KTO (Ethayarajh et al., 2024) loss to optimize the probability of *s*-MLLM for positive responses to be greater than that of *l*-MLLM and the probability of *s*-MLLM for negative responses to be lower than that of *l*-MLLM. The batch size is 256, and the learning rate is adjusted to 2e-6. Throughout all stages, we employ the adam optimizer (Diederik, 2014) and train on 16 NVIDIA A100 GPUs for one epoch each, totaling approximately 960 GPU hours. The detailed training hyperparameters are shown in Tab. 10.

Table 10: Training hyperparameters of each stage.

| Configuration | Initialization | Mimic Distillation | Preference Distillation |
|---|---|---|---|
| LLM | ✗ | ✓ | ✓ |
| VL Adaptor | ✓ | ✓ | ✓ |
| ViT | ✗ | ✗ | ✗ |
| LLM init. | Qwen-1.5-1.8B | Qwen-1.5-1.8B | 2nd-stage |
| VL Adaptor init. | MLP | 1st-stage | 2nd-stage |
| ViT init. | | CLIP-Large@336 | |
| Image resolution | | $336 \times 336$ | |
| ViT sequence length | | 576 | |
| LLM sequence length | | 2048 | |
| Optimizer | | AdamW | |
| Optimizer hyperparameter | | $\beta_1 = 0.9, \beta_2 = 0.98$ | |
| Learning rate | $1e^{-4}$ | $2e^{-5}$ | $2e^{-5}$ |
| Learning rate schedule | | Cosine decay | |
| Weight decay | | 0.0 | |
| Training epoch | | 1 | |
| Warm-up ratio | | 0.03 | |
| Global batch size | 256 | 128 | 128 |
| Numerical precision | | Bfloat16 | |
| Model parallelism | Zero2 | Zero2 offload | Zero2 offload |

Table 11: Training dataset of each stage. **#Sample** means the training samples

| Stage | Task | Dataset |
|---|---|---|
| Initialization | Captioning | LLaVA-1.5-Pretrain (Liu et al., 2023b) |
| Dense-to-Dense Distillation | Captioning | ALLaVA-Caption-4V (Chen et al., 2024), ShareGPT4V-PT (Chen et al., 2023a) |
| | Conversation | MIMIC-IT (Li et al., 2023a), LVIS (Wang et al., 2023), LRV (Liu et al., 2023a), SViT (Zhao et al., 2023) |
| Dense-to-Sparse Distillation | Captioning | ShareGPT4V-100K (Chen et al., 2023a), TextCaps (Sidorov et al., 2020) |
| | Conversation | LLaVA-1.5-Instruct (Liu et al., 2023b) |
| | General QA | GQA (Hudson & Manning, 2019), VQAv2 (Goyal et al., 2017), OKVQA (Marino et al., 2019) |
| | Grounding | Visual Genome (Krishna et al., 2017), RefCOCO (Yu et al., 2016; Mao et al., 2016) |
| | Science | AI2D (Kembhavi et al., 2016), ScienceQA (Lu et al., 2022a) |
| | Chart & Doc | DVQA (Kafle et al., 2018), ChartQA (Masry et al., 2022), DocQA (Clark & Gardner, 2018) |
| | OCR | OCRVQA (Mishra et al., 2019), SynthDoG-EN (Kim et al., 2022) |
| | Knowledge | A-OKVQA (Schwenk et al., 2022), GeoQA+ (Cao & Xiao, 2022) |
| Preference Distillation | Preference | RLAIF-V (Yu et al., 2024b) |

## A.2 TRAINING DATASETS

We curate a comprehensive open-source dataset of 5 million samples encompassing both general and expert tasks. In the Initialization stage, we utilize 0.6M general captioning samples from LLaVA-1.5-pretrain dataset (Liu et al., 2023b) to bridge the gap between visual and language modalities. In the mimic distillation stage, we utilize 2.4M general captioning and conversation samples to distill general knowledge from the teacher MLLM and 1.4M multi-tasks data samples, including VQA,

documents, science, and OCR to distill specialized knowledge from the teacher MLLM. In the preference distillation stage, we utilize 80K preference data samples to transfer preference knowledge from the teacher MLLM. The detailed dataset of each training stage is illustrated in Tab. 11.

## B  ADDITIONAL EXPERIMENTS

### B.1  MoE CONFIGURATIONS

We thoroughly explore the performance and computational trade-offs across different MoE configurations, focusing on the number of experts, the routing parameter k, and routing strategies. As shown in Tab 12, increasing the number of experts from 4 to 8 does not yield an improvement in performance. Conversely, Tab 13 illustrates that enhancing the routing parameter k from 1 to 2 results in a noticeable performance boost. This observation suggests that adjusting the routing parameter K seems to facilitate more stable training and consistent performance improvements compared to simply increasing the number of experts. One explanation for this is that adding more experts without increasing the complex knowledge of data results in each expert being insufficiently optimized. Furthermore, we investigate various routing strategies in Tab 14, comparing Top-K with RSample and Jitter. The results indicate that Top-K routing achieves the highest average performance, particularly demonstrating significant advantages in the MMB, which encompasses 20 sub-tasks. This superiority can be attributed to Top-K's ability to dynamically activate the most relevant experts based on the input, thereby showcasing its robustness across diverse tasks.

Table 12: Ablations with the expert number.

| Expert | GQA | SQA$^I$ | VQA$^T$ | MME | MMB | AVG |
|---|---|---|---|---|---|---|
| 4 | 59.3 | 68.4 | 58.7 | 68.4 | 65.1 | 64.0 |
| 8 | 58.3 | 67.4 | 58.4 | 69.1 | 64.6 | 63.6 |

Table 13: Ablations with the routing parameter k.

| Top-K | GQA | SQA$^I$ | VQA$^T$ | MME | MMB | AVG |
|---|---|---|---|---|---|---|
| 1 | 58.2 | 67.2 | 58.3 | 66.1 | 64.7 | 62.6 |
| 2 | 59.3 | 68.4 | 58.7 | 68.4 | 65.1 | 64.0 |

Table 14: Ablations with the routing strategy.

| Routing Strategy | GQA | SQA$^I$ | VQA$^T$ | MME | MMB | AVG |
|---|---|---|---|---|---|---|
| Top-K(ours) | 58.8 | 69.2 | 59.9 | 69.2 | 68.9 | 65.2 |
| RSample | 57.9 | 67.5 | 60.1 | 68.9 | 67.3 | 64.3 |
| Jitter | 58.1 | 68.4 | 60.0 | 69.1 | 67.7 | 64.6 |

### B.2  PREFERENCE-OPTIMIZATION LOSS

We explore the training stability of various preference-optimization losses. We employ KTO (Ethayarajh et al., 2024) and DPO (Rafailov et al., 2024). As shown in Tab. 15, and Tab. 16 while both methods have an impact on reducing hallucination, KTO is significantly superior to DPO. This is because KTO, which indicates whether a sample is good or bad, offers a better signal for the *s*-MLLM to surpass the *l*-MLLM compared to DPO, which determines relative quality by comparing two samples.

Table 15: Comparison between different preference-optimization loss on comprehension-oriented benchmarks.

| PO Loss | GQA | VizWiz | SQA$^I$ | VQA$^T$ | MME | MMB | MMB$^{CN}$ | AVG |
|---|---|---|---|---|---|---|---|---|
| DPO | 58.5 | 37.3 | 68.2 | 58.6 | 67.8 | 65.4 | 61.5 | 59.6 |
| KTO | 58.7 | 39.2 | 68.0 | 58.5 | 67.6 | 66.3 | 61.9 | 59.9 |

Table 16: Comparison between different preference-optimization loss on hallucination-oriented benchmarks.

| PO Loss | Object HalBench | | POPE | MMHal-Bench | |
|---|---|---|---|---|---|
| | Resp ↓ | Ment ↓ | F1 ↑ | Score ↑ | Hall ↓ |
| DPO | 20.0 | 10.5 | 86.8 | 2.68 | 18.9 |
| KTO | 11.4 | 7.2 | 87.0 | 2.76 | 17.2 |

## B.3 DISTILLATION STRATEGY

We conducted experiments to compare our method with standard Knowledge Distillation (KD) and Generalized Knowledge Distillation (GKD). To ensure a fair comparison, we utilized the same teacher and student models: a 7B dense teacher model and a 1.8B MoE student model, configured with 4 activated experts and a routing parameter k=2. The results in Tab. 17 demonstrate that our approach outperforms both KD and GKD across the comprehensive understanding benchmarks MMB and MME, as well as the hallucination benchmark Object-Halbench. This enhanced performance can be attributed to the distinctive features of our progressive distillation method, which employs a two-phase approach: it begins with dense-to-dense general distillation and is followed by dense-to-sparse multi-task distillation. This strategy facilitates a thorough assimilation of the teacher's comprehension capabilities. Moreover, our preference distillation contributes to hallucination mitigation by providing more reliable preference information derived from the robust teacher model. These advantages are absent in standard KD and GKD approaches, underscoring the additional value offered by our proposed method.

Table 17: Comparison with different distillation strategies.

| Method | MME | MMB | ObjHal |
|---|---|---|---|
| KD | 65.0 | 61.8 | 53.4 |
| GKD | 66.7 | 63.2 | 52.8 |
| MoD | 69.2 | 68.9 | 11.2 |

## B.4 TRAINING AND INFERENCE EFFICIENCY

We also conduct a detailed analysis of the training and inference costs of LLaVA-MoD-2B, with a comparison to Qwen-VL-Chat-7B in Tab. 18. Our results show that LLaVA-MoD-2B has a total of 3B parameters, with 2.2B activated during training and inference. For training costs, we report training days and the size of the dataset. LLaVA-MoD-2B achieves superior performance while using only 0.3% of the training data and 23% of the activated parameters compared to Qwen-VL-Chat-7B. For inference costs, our evaluation on a single A100-80G GPU indicates that LLaVA-MoD-2B is 2.5 faster in decoding speed, consumes 26% of FLOPs, and uses 38% of the memory compared to Qwen-VL-Chat-7B. These results demonstrate LLaVA-MoD-2B's superior efficiency while maintaining high performance.

Table 18: Comparison with training and inference costs.

| Metrics | LLaVA-MoD-2B | Qwen-VL-Chat-7B |
|---|---|---|
| Total Params (B) | 3 | 9.6 |
| Activated Params (B) | 2.2 | 9.6 |
| Dataset Size (Million) | 5 | 1450 |
| Training Time (Day) | 2 | - |
| Memory Usage (GB) | 7.8G | 20.4G |
| Inference FLOPs (TFLOPs) | 2.3 | 8.8 |
| Decoding Speed (Tokens/s) | 38.5 | 15.6 |
| Average Performance | 61.6 | 56.7 |

## C  LIMITATIONS AND FUTURE WORKS

LLaVA-MoD requires that both *s*-MLLM and *l*-MLLM belong to the same series of LLMs to ensure consistency in the vocabulary space. Future research can explore distillation techniques involving heterogeneous model families. Moreover, the requirement to load both *s*-MLLM and *l*-MLLM leads to substantial memory consumption. To enable efficient distillation, a possible solution could be to pre-extract the logits of the *l*-MLLM. This would allow only the *s*-MLLM to be loaded during training, reducing memory requirements and potentially speeding up the training process.

