# OpenReview forum: "LLaVA-MoD: Making LLaVA Tiny via MoE-Knowledge Distillation"
_ICLR.cc/2025/Conference — ICLR 2025 Poster_

### Official Review · Reviewer_uFgm · 2024-11-03

**Soundness:** 3
**Presentation:** 3
**Contribution:** 3
**Rating:** 6
**Confidence:** 4

**Summary:**

The paper proposes a framework LLaVA-MoD for distilling knowledge from large multimodal models to smaller models. The method incorporates mixture-of-experts for enhancing the expressiveness of small multimodal models. Additionally, the method uses two stage distillation -- Mimic Distillation for general knowledge alignment and Preference Distillation to enhance the student model's discriminatory capabilities for superior generalization. The experimental results are quite interesting -- The 2-billion parameter LLaVA-MoD outperforms larger models like Qwen-VL-Chat-7B by 8.8%, using only 23% of trainable parameters and 0.3% of the training data. LLaVA-MoD performs significantly well on 7 multimodal tasks and hallucination benchmarks.

**Strengths:**

The paper proposes a scalable approach for distilling large multimodal models to smaller one, which is a very relevant topic in recent time. The proposed methodology is somewhat novel and addresses key challenges faced by the existing small multimodal models. The experiments are extensive and performance improvement shown by the proposed methodology is significant.

**Weaknesses:**

1. The experimental setup is not very convincing to me. The authors claim their method to be an alternative distillation method without much empirical comparison. In order to establish the effectiveness of their method, the authors must compare some of the existing KD methods -- KD and GKD, to show why their distillation method is needed.

2. The dense-to-sparse loss is very similar to the competitive loss proposed by Sengupta et al., 2023 [1]. The authors should show how their proposed losses differ from these losses.

3. The authors should elaborately discuss why dense vs sparse architecture works. The authors mostly report the numbers with little explanation of the how part.

4. The lower performance of Mimic+preference than mimic in Table 4 should be discussed. It is counterintuitive.

5. The authors claim that their method is efficient. More detailed analysis with training/inference time and memory should be discussed.

**References**

[1] A Good Learner can Teach Better: Teacher-Student Collaborative Knowledge Distillation - ICLR'2024.

**Questions:**

See the above comments.

---

> ### Author Response · Authors · 2024-11-23
> **Response to Reviewer uFgm (1/2)**
>
> Thank you for your insightful comments. Your suggestions provide valuable guidance for refining and enhancing our work further.
>
> > **Q1: In order to establish the effectiveness of their method, the authors must compare some of the existing KD methods -- KD and GKD, to show why their distillation method is needed.**
>
> Following your valuable suggestion, we have conducted additional experiments comparing our method with standard KD and GKD. To ensure a fair comparison, we use the same teacher and student models (a 7B dense teacher model and a 1.8B MoE student model with 4 activated experts and routing parameter k=2), along with the same dataset in our paper. The results, as shown below, indicate that our approach outperforms both KD and GKD on the comprehensive understanding tasks MMB and MME, as well as on the hallucination benchmark Object-Halbench. This improved performance can be attributed to the unique features of our progressive distillation. Our mimic distillation employs a two-phase progressive approach, starting with dense-to-dense general distillation followed by dense-to-sparse multi-task distillation. This allows for comprehensive learning of the teacher's understanding capabilities. Additionally, our preference distillation enhances hallucination mitigation by providing more reliable preference information from the strong teacher model. These two advantages are not present in standard KD and GKD approaches, illustrating the added value of our proposed method.
>
>
> | **Method**      | **MME**  $\uparrow$ | **MMB**  $\uparrow$ | **Object HalBench (Resp $\downarrow$)** |
> |-------------|------|------|--------|
> | KD          | 65.0 | 61.8 |  53.4  |
> | GKD         | 66.7 | 63.2 |  52.8 |
> | **MoD (ours)**  | **69.2** |**68.9** | **11.2**  |
>
>
> ---
>
>
>
> > **Q2: The dense-to-sparse loss is very similar to the competitive loss proposed by Sengupta et al., 2023 [1]. The authors should show how their proposed losses differ from these losses.**
>
> We appreciate the valuable feedback. MPDistill, as proposed in [1], introduces a competitive loss for training a meta-teacher model aimed at enhancing **meta-learning-based** distillation. However, our approach is distinct, as it freezes the teacher model and employs progressive distillation (consisting of mimic distillation and preference distillation) to enhance the performance of the MoE student model. Therefore, our approach and MPDistill focus on distillation from fundamentally different perspectives.
>
> [1] A Good Learner Can Teach Better: Teacher-Student Collaborative Knowledge Distillation, ICLR 2024.
>
>
>
> ---
>
>
>
> > **Q3: The authors should elaborately discuss why dense vs sparse architecture works. The authors mostly report the numbers with little explanation of the how part.**
>
> We appreciate the valuable suggestion and would like to elaborate on the differences between dense and sparse architectures. The key differences lie in their approaches to model capacity and resource allocation. Dense architectures activate all model parameters uniformly for each input, which can lead to inefficiencies, particularly in handling diverse or complex tasks. Conversely, sparse architectures enhance model capacity by utilizing multiple experts and selectively activating highly relevant experts for each input. By dynamically allocating resources where they are most needed, sparse architectures combine the benefits of high model expressiveness and computational efficiency. Table 9 (main paper) demonstrates that, under the same number of activated parameters, sparse architectures achieve superior performance.
>
>
>
> ---
>
>
>
> > **Q4: The lower performance of Mimic+preference than mimic in Table 4 should be discussed. It is counterintuitive.**
>
> Thank you for your valuable concern. I would like to kindly remind the reviewer that the observed lower performance of Mimic+preference compared to Mimic alone, as shown in Table 2 (main paper), specifically occurs on comprehension benchmarks. It is important to clarify that the preference distillation primarily targets hallucination benchmarks, where we show strong results in Table 3 (main paper). This decrease in comprehension capabilities can be attributed to reward hacking [1] during preference learning. During the process, the model could exploit shortcuts, optimizing for simpler preference patterns at the expense of deeper understanding.
>
> This challenge is not unique to our method. Similar issues have been observed in MiniCPM-V and GPT-4. Several works such as KTO and DPO have been proposed to mitigate reward hacking by refining training objectives. We conducted ablation experiments with KTO and DPO, and the results in Table 16 (main paper) indicate that KTO is more robust. This suggests that incorporating KTO could enhance the balance between preference alignment and comprehension capabilities.
>
> [1] Helping or Herding? Reward Model Ensembles Mitigate but do not Eliminate Reward Hacking, COLM 2024

---

> ### Author Response · Authors · 2024-11-23
> **Response to Reviewer uFgm (2/2)**
>
> > **Q5: The authors claim that their method is efficient. More detailed analysis with training/inference time and memory should be discussed.**
>
> Thank you for your valuable suggestion and we follow up to conduct a detailed analysis of the training and inference costs of LLaVA-MoD-2B, comparing it to Qwen-VL-Chat-7B. The results, as shown below, indicate that our model achieves superior performance with less computational resources. In terms of training cost, we report on training days, trainable parameters, and dataset size. The results reveal that our model achieve superior results using merely 0.3\% of the training data and only 23\% of the trainable parameters compared to Qwen-VL-Chat-7B. For inference cost, we evaluate our model on a single A100-80G GPU with BF16 precision, reporting memory usage, inference FLOPs, and decoding speed. The results reveal that our model is approximately 2.5$\times$ faster in decoding speed, requires 26\% of FLOPs, and uses 38\% of memory compared to Qwen-VL-Chat-7B, underscoring its superior efficiency while maintaining high performance.
>
>
>
>
>
> | **Metrics**                 | **LLaVA-MoD-2B** | **Qwen-VL-Chat-7B** |
> |-------------------------|--------------|-----------------|
> | Total Params (Billion)        | 3            | 9.6             |
> | Activated Params (Billion)    | 2.2          | 9.6             |
> | Dataset Size (Million)  | 5            | 1450            |
> | Training Time (Day)     | 2            | -               |
> | Memory Usage (GB)       | 7.8          | 20.4            |
> | Inference FLOPs (TFLOPs) | 2.3          | 8.8             |
> | Decoding Speed (Tokens/s)| 38.5         | 15.6            |
> | Average Performance     | 61.6         | 56.7            |

---

> > ### Comment · Reviewer_uFgm · 2024-11-25
> > **Acknowledgment of Author Response and Further Comments**
> >
> > Thank you for your response. I will increase the score accordingly.
> >
> > Please try to add the KD baseline results to the paper. Also, if possible, compare against more unimodal KD methods.

---

> > > ### Author Response · Authors · 2024-11-26
> > >
> > > Thank you for your valuable suggestion. We will incorporate the KD baseline into our paper and explore additional KD methods.

---

### Official Review · Reviewer_vgkT · 2024-11-04

**Soundness:** 3
**Presentation:** 3
**Contribution:** 3
**Rating:** 6
**Confidence:** 3

**Summary:**

The paper introduces LLaVA-MoD, which is a framework that distills large multimodal models into efficient, small-scale ones using a sparse MoE and progressive knowledge transfer. By optimizing both model architecture and distillation strategy, LLaVA-MoD achieves high performance with minimal data and parameters, even surpassing larger models on multimodal tasks. This framework enables cost-effective deployment of high-performing models in resource-limited settings

**Strengths:**

- The paper combines MoE with knowledge distillation in a novel way, optimizing small multimodal models efficiently without sacrificing performance.
- The proposed method is robust, with a progressive distillation process that includes preference-based fine-tuning to mitigate hallucinations.
- The paper is well-written and clearly explains the framework.
- Experiments were done on a comprehensive set of evaluations.

**Weaknesses:**

- The paper does not provide an analysis of how different MoE configurations (e.g., number of experts, routing strategies, parameter k from top-k routing) impact performance and computational trade-offs. Including such an analysis could provide further valuable insights for the model.
- The results presented in Table 2 is relatively modest in improvement. LLaVA-MoD-2B only outperforms the other 2B baselines (Imp-2B) by 2.7% on average while using twice as many data samples.

**Questions:**

See above in weaknesses.

---

> ### Author Response · Authors · 2024-11-23
> **Response to Reviewer vgkT**
>
> We sincerely appreciate your helpful feedback. Your suggestions offer critical perspectives that will help us enhance our work significantly.
>
> > **Q1: The paper does not provide an analysis of how different MoE configurations (e.g., number of experts, routing strategies, parameter k from top-k routing) impact performance and computational trade-offs. Including such an analysis could provide further valuable insights for the model.**
>
>
> Thank you for the valuable suggestion. Accordingly, we thoroughly explore the performance and computational trade-offs across different MoE configurations, focusing on the number of experts, the routing parameter k, and routing strategies. Increasing the number of experts from 4 to 8, while keeping the router parameter k at 2, raises memory usage from 8G to 13G. However, the FLOPs and inference speed remain unchanged since the number of activated parameters stays the same. Increasing the routing parameter k from 1 to 2, while maintaining 4 experts, raises the FLOPs from 1.5TFlops to 2.3TFlops. However, the memory usage remains unchanged because the total number of parameters is constant.
>
> Regarding the performance, as shown in Table 13 \& 14 (main paper), adjusting the routing parameter seems to facilitate more stable training and consistent performance improvements compared to simply increasing the number of experts. One potential reason (also mentioned in reviewer k5Mn Q6) for this is that adding more experts without increasing the complex knowdege of data results in each expert being insufficiently optimized. To explore this, we conducted data scaling experiments with 8 experts. Specifically, we randomly sampled 0.7M high-quality multi-task data from the LLaVA-OneVision [1] single-image dataset and combined it with our original 1.4M multi-task data for dense-to-sparse distillation. The results show that with the inclusion of more high-quality multi-task data, the performance of the 8 experts continues to improve, highlighting the importance of scaling data alongside expert numbers for optimal performance gains.
>
> | **Data Size** | **GQA** $\uparrow$ | **SQA**^I $\uparrow$ | **VQA**^T $\uparrow$ | **MME** $\uparrow$ | **MMB** $\uparrow$ | **AVG** $\uparrow$ |
> |---------------|---------|----------------------|----------------------|---------|---------|---------|
> | 1.4M          | 58.3    | 67.4                 | 58.4                 | 69.1    | 64.6    | 63.6    |
> | **2.1M**          | **59.0**    | **68.3**                 | **58.8**                 | **71.5**    | **66.3**    | **64.8**   |
>
> Furthermore, we explore different router strategies by comparing Top-K with RSample and Jitter. These strategies do not affect computation but performance. Our results indicate that Top-K routing yields the best average performance, which implies that it is more robust across various tasks.
>
>
> | **Routing Strategy** | **GQA** $\uparrow$ | **SQA**^I $\uparrow$ | **VQA**^T $\uparrow$ | **MME** $\uparrow$ | **MMB** $\uparrow$ | **AVG** $\uparrow$ |
> |------------------|------|-----------------|-----------------|------|------|------|
> | RSample          | 57.9 | 67.5            | **60.1**            | 68.9 | 67.3 | 64.3 |
> | Jitter           | 58.1 | 68.4            | 60.0            | 69.1 | 67.7 | 64.6 |
> | **Top-K (ours)**    | **58.8** |**69.2**            | 59.9            | **69.2** | **68.9** |**65.2** |
>
> [1] LLaVA-OneVision: Easy Visual Task Transfer, Arxiv 2024
>
>
> ---
>
>
>
> > **Q2: The results presented in Table 2 is relatively modest in improvement. LLaVA-MoD-2B only outperforms the other 2B baselines (Imp-2B) by 2.7% on average while using twice as many data samples.**
>
>
> We would like to clarify that our approach employs a lower-resolution visual encoder compared to other methods, such as Imp and DeepSeek-VL, which utilize higher-resolution settings. Specifically, we use a resolution of 336, whereas Imp uses a resolution of 384, and DeepSeek-VL uses a mixture-of-resolutions of 384 \& 1024. This difference is closely related to performance. However, by leveraging 3M general caption and conversation data in our total 5M data, we are able to mitigate the limitation of the vision encoder and achieve superior performance. As shown in Table 2 (main paper), we surpass Imp-2B by 2.7\% on average. Despite the increased data utilization, our approach remains resource-efficient, encoding images into only 576 tokens compared to 729 tokens used by Imp-2B. This reduction in the number of visual tokens underscores the efficiency of our approach even with augmented data resources. Furthermore, Imp-2B reports a 3\% performance decrease when configured with the same resolution as ours, underscoring the advantage of our approach.

---

### Official Review · Reviewer_2roi · 2024-11-05

**Soundness:** 2
**Presentation:** 3
**Contribution:** 2
**Rating:** 5
**Confidence:** 4

**Summary:**

The paper introduces LLaVA-MoD, a framework aimed at distilling knowledge from large multimodal language models (MLLMs) to create smaller, efficient multimodal language models (s-MLLMs) that retain high performance with reduced computational requirements. LLaVA-MoD achieves this by employing a sparse Mixture of Experts (MoE) architecture to maintain expressive power while minimizing parameters. The distillation process is twofold: Mimic Distillation aligns the smaller model with the larger one through Kullback-Leibler (KL) divergence, and Preference Distillation enhances the smaller model’s ability to judge quality, outperforming the larger model on tasks prone to hallucination. Experimental results highlight LLaVA-MoD's efficiency, achieving significant performance improvements with minimal data and trainable parameters.

**Strengths:**

1. LLaVA-MoD achieves high performance with only a small fraction of the data and computational resources required by traditional large MLLMs. This makes it more suitable for real-world applications, especially in low-resource environments.
2. Reduced Hallucination: By incorporating preference distillation, LLaVA-MoD significantly reduces hallucination rates, making it more reliable for applications where factual accuracy is critical.
3. Flexible Architecture with MoE: The use of a sparse MoE architecture allows the model to selectively activate different "experts" for specific tasks, ensuring computational efficiency while retaining expressive power.
4. Superior Performance: LLaVA-MoD surpasses larger models like Qwen-VLChat-7B in various benchmarks, demonstrating the effectiveness of its knowledge distillation approach.

**Weaknesses:**

--The progressive distillation method, which includes dense-to-sparse transitions and multi-task training, could be complex to implement and tune effectively.
--While preference distillation enhances hallucination mitigation, it does not consistently improve the model’s comprehension capabilities, which might limit its applicability in understanding complex instructions.
--The effectiveness of LLaVA-MoD is heavily reliant on the quality and capacity of the teacher model. If the teacher model is limited, the distilled s-MLLM might inherit those limitations.

**Questions:**

--What's the computational complexity?
--How do you address the limitation of the teacher's model?

---

> ### Author Response · Authors · 2024-11-23
> **Response to Reviewer 2roi (1/2)**
>
> Thank you for your constructive feedbacks. Your insights are very helpful in making our work even better.
>
>
> >**Q1: The progressive distillation method, which includes dense-to-sparse transitions and multi-task training, could be complex to implement and tune effectively**
>
> We appreciate your constructive concern regarding the complexity of the progressive distillation. While our approach may introduce additional complexities, the results in Table 2 (main paper) demonstrate the superior performance. We are committed to providing comprehensive documentation and guidance to help facilitate implementation and tuning.
>
> Moreover, we would like to clarify that the rationale behind our progressive distillation approach, which includes dense-to-dense, dense-to-sparse, and preference distillation, is to ensure both effectiveness and stability throughout the knowledge transfer process. Initially, we employ dense-to-dense distillation to create a single expert capable of capturing general knowledge. This initial step is crucial because jumping straight to dense-to-sparse distillation often leads to instability in MoE training and can cause model collapse. By building a solid foundation with dense-to-dense distillation, we can create a mixture of general experts, which helps us later derive specialized experts through dense-to-sparse distillation. Our experiments, as presented in Table 7 (main paper), demonstrate that the progressive approach outperforms direct dense-to-sparse distillation. Furthermore, preference distillation is specifically aimed at reducing hallucinations, addressing a significant challenge in model performance.
>
>
>
> ---
>
>
>
> > **Q2: While preference distillation enhances hallucination mitigation, it does not consistently improve the model’s comprehension capabilities, which might limit its applicability in understanding complex instructions.**
>
>
> We appreciate your valuable feedback regarding the relationship between preference distillation and the model's comprehension capabilities. We would like to clarify that the primary goal of preference distillation is to reduce hallucinations, rather than to improve understanding of complex instructions.
>
> While addressing hallucinations, preference distillation, or more broadly, preference learning, faces the challenge of reward hacking [1], where focusing too much on mitigating hallucination could degrade the model's general capabilities. This phenomenon has been observed in prior works such as GPT-4 and MiniCPM-V, and many works explore strategies to strike a balance, ensuring that the model can both mitigate hallucinations and maintain or improve their understanding of complex instructions.
>
> We meticulously adjusted key parameters, including a small learning rate of 2e-6, fewer trainable parameters in the MoE architecture, and a stable KTO loss, to achieve this balance. The results presented in Tables 4 and 5 (main paper) demonstrate that our approach significantly reduces hallucinations while maintaining an average decrease of only 0.3\% in comprehension capabilities across seven benchmarks, which is an acceptable trade-off.
>
>
> [1] Helping or Herding? Reward Model Ensembles Mitigate but do not Eliminate Reward Hacking, COLM 2024
>
>
> ---
>
>
>
> > **Q3: The effectiveness of LLaVA-MoD is heavily reliant on the quality and capacity of the teacher model. If the teacher model is limited, the distilled s-MLLM might inherit those limitations. How do you address the limitation of the teacher's model?**
>
>
> Thank you for your insightful suggestion about the limitations of the teacher model. As shown in Table 1 \& Table 3 (main paper), the teacher model performs well in comprehension but poorly in mitigating hallucinations. We propose mimic distillation to distill the strong comprehension capabilities from the teacher model and propose preference distillation to address the teacher model's inherent limitations in mitigating hallucinations. Our results in Table 3 demonstrate that our approach achieves superior performance compared to RLHF-based methods and even surpasses that of the teacher model.

---

> ### Author Response · Authors · 2024-11-23
> **Response to Reviewer 2roi (2/2)**
>
> >**Q4: What's the computational complexity?**
>
> We appreciate your suggestion about the computational complexity and discuss the computational complexity of LLaVA-MoD-2B by analyzing both the training and inference phases compared to Qwen-VL-Chat-7B. Regarding training complexity, we report the training duration, number of trainable parameters, and dataset size, highlighting that LLaVA-MoD-2B achieves superior performance with only 0.3\% of the training data and 23\% of the trainable parameters compared to Qwen-VL-Chat-7B. Regarding inference complexity, we evaluate it on a single A100-80G GPU loaded in BF16 precision, reporting memory usage, inference FLOPs, and decoding speed. The results show that LLaVA-MoD-2B is approximately 2.5 times faster in decoding speed, requiring 26\% of FLOPs and 38\% of memory compared to Qwen-VL-Chat-7B, thereby demonstrating its superior efficiency while maintaining high performance.
>
>
> | **Metrics**                 | **LLaVA-MoD-2B** | **Qwen-VL-Chat-7B** |
> |-------------------------|--------------|-----------------|
> | Total Params (Billion)        | 3            | 9.6             |
> | Activated Params (Billion)    | 2.2          | 9.6             |
> | Dataset Size (Million)  | 5            | 1450            |
> | Training Time (Day)     | 2            | -               |
> | Memory Usage (GB)       | 7.8          | 20.4            |
> | Inference FLOPs (TFLOPs) | 2.3          | 8.8             |
> | Decoding Speed (Tokens/s)| 38.5         | 15.6            |
> | Average Performance     | 61.6         | 56.7            |

---

### Official Review · Reviewer_EM2J · 2024-11-05

**Soundness:** 4
**Presentation:** 3
**Contribution:** 3
**Rating:** 8
**Confidence:** 3

**Summary:**

This work introduces a framework called LLaVA-MoD for building efficient small-scale multimodal large language models (MLLMs) using a sparse Mixture-of-Experts (MoE) architecture. The framework specifically applies MoE to MLP layers within transformer blocks. To preserve the capabilities of larger MLLMs while maintaining efficiency, the authors developed a specialized distillation approach combining KL divergence loss and Preference Optimization (PO) loss between the student (small) and teacher (large) models.

**Strengths:**

This work introduces a new framework for better tuning small-scale MLLM. This framework provides solutions in real world problems by offering a more efficient approach to model inference and deployment across diverse application scenarios.

The experiment is comprehensive, covering a wide range of tasks and models, on both general comprehension benchmarks and hallucination benchmarks. The design of s-MLLM is efficient and effective, achieving superior performance compared to other models of similar size across multiple benchmark tests. The distillation pipeline is clear and well defined. The paper is clearly written.

**Weaknesses:**

This work  can also be related to inference efficiency and inference acceleration. It would be good if it included inference measurements like FLOPs and inference latency showing the work’s practical benefits in deployment scenarios.

**Questions:**

1. There are more small-scale MLLM coming out these days with higher performance and comparable size (0.5B) [1]. Could the sparse MoD architecture and distillation strategy be applied to these newer models to further enhance their performance?

2. I am wondering about the performance of Preference Distillation vs DPO in the second stage. Does it show advantages using distillation instead of direct DPO, in terms of model performance or training efficiency?


[1] Li, Bo, et al. "Llava-onevision: Easy visual task transfer." arXiv preprint arXiv:2408.03326 (2024).

---

> ### Author Response · Authors · 2024-11-23
> **Response to Reviewer EM2J**
>
> We are very grateful for your thoughtful feedback. Your guidance is crucial in advancing our work.
>
> >**Q1: This work can also be related to inference efficiency and inference acceleration. It would be good if it included inference measurements like FLOPs and inference latency showing the work’s practical benefits in deployment scenarios.**
>
>
> We appreciate your suggestion regarding inference costs. We follow up to conduct a detailed evaluation of LLaVA-MoD-2B and compare it with Qwen-VL-Chat-7B. The evaluation is performed on a single A100-80G GPU, with both models loaded in BF16 precision. We present several key metrics, including memory usage, inference FLOPs, and decoding speed. The results show that LLaVA-MoD-2B has a decoding speed of approximately 2.5 times faster than Qwen-VL-Chat-7B. Additionally, it consumes 26\% of FLOPs and 38\% of memory compared to Qwen-VL-Chat-7B, demonstrating its efficiency while maintaining high performance.
>
>
> | **Metrics**                 | **LLaVA-MoD-2B** | **Qwen-VL-Chat-7B** |
> |-------------------------|--------------|-----------------|
> | Total Params (Billion)        | 3            | 9.6             |
> | Activated Params (Billion)    | 2.2          | 9.6             |
> | Dataset Size (Million)  | 5            | 1450            |
> | Training Time (Day)     | 2            | -               |
> | Memory Usage (GB)       | 7.8          | 20.4            |
> | Inference FLOPs (TFLOPs) | 2.3          | 8.8             |
> | Decoding Speed (Tokens/s)| 38.5         | 15.6            |
> | Average Performance     | 61.6         | 56.7            |
>
>
>
> ---
>
>
>
> > **Q2: There are more small-scale MLLM coming out these days with higher performance and comparable size such as LLaVA-OneVision-0.5B. Could the sparse MoD architecture and distillation strategy be applied to these newer models to further enhance their performance?**
>
> We follow the insightful suggestion to explore the effectiveness of our approach on LLaVA-OneVision. We use LLaVA-OneVision-7B(SI) as the teacher model, and LLaVA-OneVision-0.5B(SI) as the student model. We train on the single image dataset, where the original dataset in stage-1 and stage-2 are used for dense-to-dense distillation, and the original dataset in stage-2 is used for dense-to-sparse distillation. We equip the LLM in LLaVA-OV-0.5B(SI) with 4 experts and employ Top-2 routing strategy during training and inference. The results below show that our MoE distillation can further enhance the performance of LLaVA-OneVision, demonstrating the generalization on various small-scale MLLMs.
>
>
>
> | **Method**      | **AI2D** $\uparrow$ | **ChartQA** $\uparrow$ | **MathVerse** $\uparrow$ | **MathVista** $\uparrow$ | **MMB** $\uparrow$  | **MME** $\uparrow$  | **AVG** $\uparrow$ |
> |-------------|------|---------|-----------|-----------|------|------|------
> | Original    | 54.2 | 61.0    | 17.3      | 34.6      | 43.8 | 60.9 |  45.3  |
> | **MoD (ours)** | **56.3** | **61.7**    | **18.1**      | **36.1**     | **47.4** | **63.5** |  **47.2**   |
>
>
>
> ---
>
>
>
> >**Q3: I am wondering about the performance of Preference Distillation vs DPO in the second stage. Does it show advantages using distillation instead of direct DPO, in terms of model performance or training efficiency?**
>
>
> Thank you for your valuable feedback. We follow up to conduct experiments to compare the performance of Preference Distillation and direct DPO on LLaVA-MoD-2B. The results indicate that Preference Distillation is more effective in mitigating hallucinations. This advantage comes from the strong teacher model, which provides more reliable samples compared to the student’s own outputs during DPO. Specifically, the student model is required to score higher on positive samples and lower on negative samples relative to the teacher model. This enhances the student’s learning process and leads to more effective hallucination mitigation.
>
>
>
> | **Method**                   | **Object HalBench** |          | **POPE**  | **MMHal-Bench**  |           |
> |--------------------------|-----------------|----------|-------|--------------|-----------|
> |                          | **Resp** $\downarrow$     | **Ment** $\downarrow$ | **F1** $\uparrow$ | **Score** $\uparrow$   | **Hall** $\downarrow$ |
> | Direct DPO               | 22.5            | 16.2     | 86.7  | 2.65         | 18.7      |
> | **Preference Distillation**  | **11.4**        | **7.2**  | **87.0** | **2.76**     | **17.2**   |

---

### Official Review · Reviewer_vZAr · 2024-11-06

**Soundness:** 3
**Presentation:** 3
**Contribution:** 3
**Rating:** 6
**Confidence:** 3

**Summary:**

This paper presents LLaVA-MoD, a framework for efficiently training a small-scale multimodal language model (MLLM) by distilling knowledge from a larger, more capable MLLM (I'll denote this as L for brevity). The main contributions are twofold: introducing a sparse MoE architecture for the student (I'll denote this as S for brevity) and developing a progressive distillation strategy. The MoE architecture adds multiple FFN heads with a linear routing mechanism, for efficiency and model expressiveness. The progressive distillation approach occurs in two stages: (1) mimic distillation, which minimizes KL divergence to align S’s outputs with the L’s outputs, and (2) preference distillation, which enhances the student’s ability to distinguish preference between examples, thus reducing hallucination rates. Compared to existing approaches, LLaVA-MoD is notably efficient both in terms of training data and the number of trainable parameters.

**Strengths:**

- The paper provides extensive experimental comparisons with prior work, especially highlighting gains in data efficiency and trainable parameters, which is a strong advantage for low-resource training.
- I found the preference distillation interesting in that it reduces hallucination. This aspect could benefit from a deeper explanation of the mechanism behind this reduction, but it’s a notable strength.
- Given the complexity of the architecture, the authors include an extensive ablation study to evaluate each component’s effectiveness, which helps to validate the design choices.

**Weaknesses:**

- While training sample efficiency and the number of trainable parameters are discussed, it would be helpful to see more detail on the total parameters required for training and inference. Efficiency in trainable parameters doesn’t always translate directly to lower memory or computational costs-full backpropagation through the LLM is still necessary even if only adapter and FFN heads are updated. Information on actual memory usage or wall-clock time would give a more complete picture of the framework’s practical efficiency.
- The architecture is fairly complex, and although the authors do a good job with the explanation, some details remain a bit unclear. For example, the architectural differences between S and L could be more explicitly described to clarify distinctions. See questions

**Questions:**

- Since the paper emphasizes data efficiency, could you clarify how the other methods in the comparison were trained? For instance, is there any early-stopping strategy, or specific training limits set for them? And for LLaVA, how do you know that “5M” samples are enough?
Do the S and L need to be the same architecture, or is there some flexibility here? As I understand it, L is distinct from S, where S is initially trained as a dense model before introducing sparsity. Is that correct?
- For Table 1, could you clarify the comparison parameters? Are the final model sizes (S) similar across comparisons? And do the models differ in their teacher models, or is it consistent in that regard?

---

> ### Author Response · Authors · 2024-11-23
> **Response to Reviewer vZAr (1/2)**
>
> We sincerely value your insightful comments. Your advice significantly helps in enhancing the quality.
>
> >**Q1: While training sample efficiency and the number of trainable parameters are discussed, it would be helpful to see more detail on the total parameters required for training and inference. Efficiency in trainable parameters doesn’t always translate directly to lower memory or computational costs-full backpropagation through the LLM is still necessary even if only adapter and FFN heads are updated. Information on actual memory usage or wall-clock time would give a more complete picture of the framework’s practical efficiency.**
>
> Thank you for your valuable suggestion. In our approach, only a subset of experts are activated during both the training and inference. This means that the number of parameters involved in computation remains consistent across both stages. This design ensures efficient resource utilization, optimizing both the computational efficiency and performance of the model.
>
> We also conduct a detailed analysis of the training and inference costs for LLaVA-MoD-2B, with a comparison to Qwen-VL-Chat-7B. Our results show that LLaVA-MoD-2B has a total of 3B parameters, with 2.2B activated during the training and inference. For training costs, we report on training days and dataset size. LLaVA-MoD-2B achieves superior performance while using only 0.3\% of the training data and 23\% of the activated parameters compared to Qwen-VL-Chat-7B. For inference costs, our evaluation on a single A100-80G GPU indicates that LLaVA-MoD-2B is 2.5$\times$ faster in decoding speed, consumes 26% of the FLOPs, and uses 38% of the memory compared to Qwen-VL-Chat-7B. These results demonstrate LLaVA-MoD-2B's superior efficiency while maintaining high performance.
>
> | **Metrics**                 | **LLaVA-MoD-2B** | **Qwen-VL-Chat-7B** |
> |-------------------------|--------------|-----------------|
> | Total Params (Billion)        | 3            | 9.6             |
> | Activated Params (Billion)    | 2.2          | 9.6             |
> | Dataset Size (Million)  | 5            | 1450            |
> | Training Time (Day)     | 2            | -               |
> | Memory Usage (GB)       | 7.8          | 20.4            |
> | Inference FLOPs (TFLOPs) | 2.3          | 8.8             |
> | Decoding Speed (Tokens/s)| 38.5         | 15.6            |
> | Average Performance     | 61.6         | 56.7            |
>
>
>
> ---
>
>
>
> >**Q2: The architecture is fairly complex, and although the authors do a good job with the explanation, some details remain a bit unclear. For example, the architectural differences between S and L could be more explicitly described to clarify distinctions.**
>
>
> Thank you for your valuable feedback. We would like to clarify the implementation details provided in Section 4.1 regarding our model architecture. Both the student and teacher models follow the ViT-MLP-LLM architecture. The key differences between the student and teacher models primarily lie in their LLMs. The teacher model employs a 7B size dense LLM, while the student model employs a 2B size LLM that incorporates two different architectures. The student is first initialized to a dense architecture for dense-to-dense distillation, and subsequently, its feedforward network (FFN) is transformed into a sparse MoE structure through sparse upcycling, allowing for both dense-to-sparse distillation and preference distillation. Ultimately, the student model utilizes a sparse MoE architecture.
>
>
> ---

---

> ### Author Response · Authors · 2024-11-23
> **Response to Reviewer vZAr (2/2)**
>
> >**Q3: Since the paper emphasizes data efficiency, could you clarify how the other methods in the comparison were trained? For instance, is there any early-stopping strategy, or specific training limits set for them? And for LLaVA, how do you know that “5M” samples are enough?**
>
>
> Thank you for your valuable feedback.  We show data size of other methods in Table 2 (main paper). For example, Qwen-VL-Chat-7B model uses 1450M data and DeepSeek-VL-1.3B model uses 2000M data. These methods do not use any early-stopping strategy or specific training limits.
>
>
> We would like to clarify that our approach focuses more on training efficiency, aiming to achieve superior performance with fewer data and reduced computational resources. The 5M training samples we used are sourced from publicly available datasets. To maximize their utility, we employ progressive distillation, which leverages different subsets of data at various stages of training. This approach enables the model to learn progressively and make the most of the available data. As a result, our method achieves an 8.8\% performance improvement over Qwen-VL-Chat-7B while using only 0.3\% of training data and 23\% of trainable parameters. These results highlight the efficiency and effectiveness of our approach.
>
>
> ---
>
>
>
> >**Q4: Do the S and L need to be the same architecture, or is there some flexibility here? As I understand it, L is distinct from S, where S is initially trained as a dense model before introducing sparsity. Is that correct?**
>
>
> We appreciate the question and would like to clarify that the student and teacher models do not need to be the same structure. Both the student and teacher models utilize the same vision encoder and VL adaptor. However, their LLM components differ: the student model is initially trained as a dense structure and later transformed into a sparse MoE structure through sparse upcycling, whereas the teacher model remains a dense structure throughout the process. This flexibility allows for architectural differences between the student and teacher models while maintaining compatibility for knowledge transfer.
>
> ---
>
> >**Q5: For Table 1, could you clarify the comparison parameters? Are the final model sizes (S) similar across comparisons? And do the models differ in their teacher models, or is it consistent in that regard?**
>
>
> Thank you for your valuable questions. Table 1 (main paper) shows the performance of the teacher models of two different series of LLMs used for distillation. The student models utilize the corresponding two series of LLMs. The student models differ from their teacher models in both scale and architecture: the students have 2B parameters and adopt a sparse MoE architecture, whereas the teacher models have 7B parameters and use a dense architecture. To ensure fairness and consistency in the evaluation, all student models being compared maintain the same scale of 2B parameters across the two series.

---

### Official Review · Reviewer_k5Mn · 2024-11-10

**Soundness:** 3
**Presentation:** 3
**Contribution:** 2
**Rating:** 8
**Confidence:** 3

**Summary:**

LLaVA-MoD introduces a framework for creating efficient small-scale multimodal language models through knowledge distillation from larger models. The approach tackles two key challenges: optimizing network structure through sparse Mixture of Experts (MoE) architecture, and implementing a progressive knowledge transfer strategy. This strategy combines mimic distillation, which transfers general and specialized knowledge, with preference distillation to reduce hallucinations. The framework represents an advancement in making multimodal language models more efficient and practical for real-world applications.

**Strengths:**

The technical design features a well-constructed progressive distillation strategy that combines mimic distillation for knowledge transfer with preference distillation for reducing hallucinations. The integration of MoE architecture into small-scale models is thoughtfully implemented, striking an effective balance between computational efficiency and model performance.

The empirical validation is thorough. The authors present comprehensive evaluations across multiple benchmarks, showing that LLaVA-MoD-2B not only outperforms larger models while using fewer resources, but also achieves better results in hallucination reduction compared to existing methods. The ablation studies demonstrate the contribution of each component to the overall performance.

The work also shows strong practical relevance, addressing the challenge of deploying efficient multimodal language models in resource-constrained environments. The reduction in required training data and parameters while maintaining or improving performance makes this work valuable for real-world applications.

**Weaknesses:**

The most significant technical constraint is the requirement that student and teacher models must belong to the same LLM family to ensure vocabulary space consistency. This limitation restricts the framework's flexibility and broader applicability. Additionally, the training process demands substantial memory resources as it requires loading both student and teacher models simultaneously.

The experimental evaluation, while comprehensive in many aspects, lacks detailed analysis in key areas. The paper could benefit from a more thorough examination of inference costs and computational requirements in real-world deployment scenarios. There is also limited exploration of training stability across different configurations and minimal discussion of failure cases or error patterns.

The architectural exploration could be more extensive, particularly regarding different MoE configurations and alternative approaches to reducing memory requirements during training.

**Questions:**

Why does increasing the number of experts from 4 to 8 fail to improve performance, and is there a fundamental limitation in the MoE architecture?

How can the framework be modified to enable distillation between different model families, rather than requiring the same LLM family for student and teacher?

What specific mechanisms enable preference distillation to reduce hallucinations more effectively than RLHF-based methods?

---

> ### Author Response · Authors · 2024-11-23
> **Response to Reviewer k5Mn (1/3)**
>
> Thanks a lot for your valuable comments. Your suggestions are very helpful in further improving the work.
>
> > **Q1: The student and teacher models must belong to the same LLM family to ensure vocabulary space consistency. How can the framework be modified to enable distillation between different model families, rather than requiring the same LLM family for student and teacher?**
>
> We clarify that our approach does not require the teacher and student models to share the same LLM architecture. However, their vocabularies must align, including having the same size and token IDs. Given the teacher output with dimension ($N$, $V^T$) and the student output with dimension ($N$, $V^S$), where $N$ represents the number of tokens, $V^T$ and $V^S$ represent the vocabulary size, our approach minimizes the KL divergence between these two outputs in the vocabulary dimension to achieve precise token-level knowledge transfer. This means that the vocabulary must have the same size and the same token IDs, which prevents ambiguity in the distillation. Thus, this constraint is essential to ensure accurate and efficient knowledge transfer while allowing flexibility in model architecture.
>
> Recent work [1] in the area of LLMs has tried to address this challenge through ensemble vocabulary alignment. However, to our knowledge, similar approaches have not yet been explored within the area of MLLMs. We will actively explore vocabulary alignment between heterogeneous MLLMs in our future work.
>
> [1] Bridging the Gap between Different Vocabularies for LLM Ensemble, ACL 2024
>
>
> ---
>
>
> > **Q2: Additionally, the training process demands substantial memory resources as it requires loading both student and teacher models simultaneously.**
>
> We acknowledge that in the era of large models, the distillation process requiring simultaneous loading of student and teacher models can be memory-intensive. To address this issue, we employ BF16 precision, flash attention, and DeepSpeed's ZeRO-2 optimization to significantly reduce memory usage and enables efficient training of a 7B teacher distilling a 2B student on 16 A100-80G GPUs. Moreover, as the teacher model is frozen, we can pre-extract the teacher's logit outputs. This eliminates the need to load the teacher model during training, thereby reducing memory consumption. With this approach, the memory requirements during distillation become comparable to those for fine-tuning the student model alone.
>
>
> ---
>
>
> > **Q3: The experimental evaluation, while comprehensive in many aspects, lacks detailed analysis in key areas. The paper could benefit from a more thorough examination of inference costs and computational requirements in real-world deployment scenarios.**
>
> Thanks for your valuable suggestion. We provide a detailed analysis of the inference costs regarding LLaVA-MoD-2B and a comparative evaluation against Qwen-VL-Chat-7B. The evaluation is conducted on a single A100-80G GPU with the model loaded in BF16 precision. The results show that LLaVA-MoD-2B achieves approximately 2.5$\times$ faster decoding speed, consumes 26\% of FLOPs, and uses 38\% of memory usage compared to Qwen-VL-Chat-7B. These findings demonstrate the superior efficiency of LLaVA-MoD-2B while maintaining high performance, making it well-suited for real-world deployment scenarios.
>
> | **Metrics**                 | **LLaVA-MoD-2B** | **Qwen-VL-Chat-7B** |
> |-------------------------|--------------|-----------------|
> | Total Params (Billion)        | 3            | 9.6             |
> | Activated Params (Billion)    | 2.2          | 9.6             |
> | Dataset Size (Million)  | 5            | 1450            |
> | Training Time (Day)     | 2            | -               |
> | Memory Usage (GB)       | 7.8          | 20.4            |
> | Inference FLOPs (TFLOPs) | 2.3          | 8.8             |
> | Decoding Speed (Tokens/s)| 38.5         | 15.6            |
> | Average Performance     | 61.6         | 56.7            |

---

> ### Author Response · Authors · 2024-11-23
> **Response to Reviewer k5Mn (2/3)**
>
> > **Q4: The architectural exploration could be more extensive, particularly regarding different MoE configurations and alternative approaches to reducing memory requirements during training.**
>
> Thank you for your valuable suggestion. We have conducted some architectural explorations, including the number of experts and parameter K from Top-K routing, which can be found in Appendix B.1. The finding is that adjusting the routing parameter K seems to facilitate more stable training and consistent performance improvements compared to simply increasing the number of experts. Additionally, we explore different routing strategies here. We compare Top-K with RSample and Jitter. The results below show that Top-K routing yields the best average performance. In particular, Top-K routing demonstrates significant advantages on the MMB which contains 20 sub-tasks. This superiority could be attributed to its ability to dynamically activate the most relevant experts based on each input, indicating robustness across various tasks.
>
>
> | **Routing Strategy** | **GQA** $\uparrow$ | **SQA**^I $\uparrow$ | **VQA**^T $\uparrow$ | **MME** $\uparrow$ | **MMB** $\uparrow$ | **AVG** $\uparrow$ |
> |------------------|------|-----------------|-----------------|------|------|------|
> | RSample          | 57.9 | 67.5            | **60.1**            | 68.9 | 67.3 | 64.3 |
> | Jitter           | 58.1 | 68.4            | 60.0            | 69.1 | 67.7 | 64.6 |
> | **Top-K (ours)**    | **58.8** |**69.2**            | 59.9            | **69.2** | **68.9** |**65.2** |
>
>
>
> We have employed BF16 precision, flash attention, and DeepSpeed's ZeRO-2 optimization to reduce memory usage during training. Additionally, as an alternative approach, we can pre-extract the teacher’s logit output, eliminating the need to load the teacher model during training. In this case, the memory consumption during distillation is comparable to that required for fine-tuning the student model only.
>
> ---
>
>
>
> > **Q5: There is also limited exploration of training stability across different configurations and minimal discussion of failure cases or error patterns.**
>
> We appreciate your insightful cocern about training stability. As detailed in our paper, we have conducted analysis in this regard, and we would like to summarize them here for clarity. The impact of the model architecture on training stability is explored in Section 4.3.3 and Appendix B.1. Table 13 (main paper) demonstrates that increasing the number of experts can introduce training difficulties. The impact of the distillation loss on training stability is also examined in Section 4.3.2 and Appendix B.2. We compare the robustness of different distillation strategies, illustrating that KTO provides greater stability than DPO during preference distillation. We further compare the stability of routing strategy as mentioned in the response to Q4 for a comprehensive exploration.
>
> Our method shows relatively lower performance on certain high-resolution image benchmarks, such as OCR. This is because we utilize CLIP with a resolution of 336 as the vision encoder, and the base resolution may limit the model’s ability to process fine-grained details in high-resolution images. We will include a detailed discussion of these failure cases and their potential mitigations in the revised version.

---

> ### Author Response · Authors · 2024-11-23
> **Response to Reviewer k5Mn (3/3)**
>
> > **Q6: Why does increasing the number of experts from 4 to 8 fail to improve performance, and is there a fundamental limitation in the MoE architecture?**
>
> Thank you for your insightful question. Increasing the number of experts reduces the activation probability for each expert, making it challenging for the router to effectively assign tokens to the most suitable experts, resulting in imbalanced expert
> activation. This imbalance leads to insufficient training for some experts and limits overall performance [1,2,3]. The scalability of MoE architectures depends heavily on the dataset size relative to the number of experts. For example, DeepSeekMoE [4] successfully trains 64 experts using billion-scale datasets, demonstrating the importance of matching dataset size to model capacity to ensure balanced expert training.
>
> To validate this, we conduct additional experiments with 8 experts. We augment the training data by adding 0.7M high-quality multi-task samples from the LLaVA-OneVision [5] single-image dataset to our original 1.4M dataset for dense-to-sparse distillation. The results show consistent performance improvements with the larger dataset, confirming that both data quality and quantity are crucial for effectively scaling the number of experts.
> These findings highlight the potential of our approach and underscore the importance of optimizing the data-to-expert ratio for future improvements.
>
> | **Data Size** | **GQA** $\uparrow$ | **SQA**^I $\uparrow$ | **VQA**^T $\uparrow$ | **MME** $\uparrow$ | **MMB** $\uparrow$ | **AVG** $\uparrow$ |
> |---------------|---------|----------------------|----------------------|---------|---------|---------|
> | 1.4M          | 58.3    | 67.4                 | 58.4                 | 69.1    | 64.6    | 63.6    |
> | **2.1M**          | **59.0**    | **68.3**                 | **58.8**                 | **71.5**    | **66.3**    | **64.8**   |
>
>
> [1] Switch Transformers: Scaling to Trillion Parameter Models with Simple and Efficient Sparsity, JMLR 2022.
>
> [2] Outrageously Large Neural Networks: The Sparsely-Gated Mixture-of-Experts Layer, ICLR 2017.
>
> [3] GShard: Scaling Giant Models with Conditional Computation and Automatic Sharding, ICLR 2021.
>
> [4] DeepSeekMoE: Towards Ultimate Expert Specialization in Mixture-of-Experts Language Models, Arxiv 2024
>
> [5] LLaVA-OneVision: Easy Visual Task Transfer, Arxiv 2024
>
>
>
> ---
>
>
>
> > **Q7: What specific mechanisms enable preference distillation to reduce hallucinations more effectively than RLHF-based methods?**
>
>
> The key distinction between preference distillation and RLHF-based methods lies in which model acts as the reference model. RLHF-based methods employ itself as the reference model, which can limit the quality of training signals and result in residual hallucinations. Instead, preference distillation employs a strong teacher model as the reference model, providing high-quality positive and negative samples. The student model learns to score higher on positive samples and lower on negative ones, leading to more effective mitigation of hallucinations through reliable external guidance. To validate preference distillation's effectiveness, we conduct experiments comparing it with DPO on LLaVA-MoD-2B, demonstrating its superior performance in reducing hallucinations, as detailed below.
>
>
> | **Method**                   | **Object HalBench** |          | **POPE**  | **MMHal-Bench**  |           |
> |--------------------------|-----------------|----------|-------|--------------|-----------|
> |                          | **Resp** $\downarrow$     | **Ment** $\downarrow$ | **F1** $\uparrow$ | **Score** $\uparrow$   | **Hall** $\downarrow$ |
> | Direct DPO               | 22.5            | 16.2     | 86.7  | 2.65         | 18.7      |
> | **Preference Distillation**  | **11.4**        | **7.2**  | **87.0** | **2.76**     | **17.2**   |

---

> > ### Comment · Reviewer_k5Mn · 2024-11-25
> >
> > Thank the authors for the thorough reply. My comments and questions have been addressed, and I increased the score.

---

> > > ### Author Response · Authors · 2024-11-26
> > >
> > > Thank you for your time and effort. Your valuable feedback has significantly improved the quality of our work

---

### Meta-Review · Area_Chair_GC77 · 2024-12-19

**Metareview:**

This work introduces LLaVA-MoD, a framework for training compact multimodal language models through knowledge distillation from larger models. The study addresses two main challenges: (1) optimizing the network structure using a sparse Mixture of Experts (MoE) architecture, and (2) implementing a progressive knowledge transfer strategy to acquire both general and specialized knowledge via preference optimization / alignment. The experimental results effectively demonstrate the benefits of the proposed framework.

Overall, five out of six reviewers recognize the soundness and contributions of the proposed techniques, as well as the experimental improvements. Only one reviewer raised concerns regarding the complexity of the techniques and the limitations inherited from the teacher model. The authors have committed to providing detailed documentation and guidelines to facilitate implementation and tuning, which addresses the concern about complexity.

Regarding the limitations inherited from the teacher model, the authors introduced mimic distillation to capture the strong comprehension capabilities of the teacher model and preference distillation to address the teacher model’s inherent limitations in mitigating hallucinations. These efforts are partially validated by Table 3, where the student model even outperforms the teacher model in certain metrics.

Given these contributions and the steps taken to address the identified weaknesses, we conclude that the strengths of the work outweigh its limitations, many of which have been partially addressed. Therefore, we recommend accepting this submission. But for authors, please follow the reviewers' feedback and your promise to revise the paper.

**Additional Comments On Reviewer Discussion:**

I mainly list the key concerns from Reviewer 2roi which gives the only negative score.
1)	the complexity of the techniques: dense-to-sparse transitions and multi-task training could be complex to implement and tune effectively.
The authors have committed to providing detailed documentation and guidelines to facilitate implementation and tuning, which addresses the concern about complexity.
2)	the limitations inherited from the teacher model.
the authors introduced mimic distillation to capture the strong comprehension capabilities of the teacher model and preference distillation to address the teacher model’s inherent limitations in mitigating hallucinations. These efforts are partially validated by Table 3, where the student model even outperforms the teacher model in certain metrics.
I think the authors address these issues, and thus agree with the authors. Moreover, most reviewers do not have these concerns.

---

### Decision · Program_Chairs · 2025-01-22

Accept (Poster)